# Highly replicating hepatitis C virus variants emerge in immunosuppressed patients causing severe disease

Paul Rothhaar [1], Tomke Arand[1], Ha Gyu-Thomas Seong[1,2], Christian Heuss[1], Margaret Tulessin [3], Zhiqing Wang[4,5], Colin Förster [1], Alina C. Schneider[1], Jocelyn Quistrebert [4], Haiting Chai[4], Marvin Reineke [6], Louise Benning [6], Jonathan Honegger[7,8], Maike Hofmann [9], Robert Thimme [9], Jörg Timm [10], STOPHCV investigators*, Paul Schnitzler[11], Uta Merle [12], Naglaa H. Shoukry [13,14], Julie Bruneau[13,15], Chaturaka Rodrigo [16,17], Andrew Lloyd [16], Rowena A. Bull [16,17], M. Azim Ansari[4], Carolin Mogler [3], John McLauchlan[18], Xavier Forns[19], Sofía Pérez-del-Pulgar[19] & Volker Lohmann [1,2] ✉

Hepatitis C virus (HCV) exists as a heterogenous quasispecies, but the phenotypic consequences of viral variability are widely unexplored. Here we identify a replication enhancing domain (ReED) in non-structural protein 5A conferring high replication fitness to clinical isolates. Accumulation of mutations in the ReED mediates high genome replication capacity. In a cohort of liver transplant patients, high replicator variants are exclusively found in individuals with severe disease outcome, suggesting that high viral replication fitness is associated with increased viral pathogenesis. Analysis of large sequence cohorts reveals that overall only 10% of viral genomes show genetic signatures of high replicators, which are enriched in recipients of liver transplantations, patients developing hepatocellular carcinoma and in HIV coinfected individuals. Overall, our data suggests that low replication fitness is a hallmark of HCV, contributing to establishment of persistence, whereas high replicators appear to have an advantage under conditions of immune suppression, thereby enforcing pathogenesis.

Infections with the hepatitis C virus (HCV) become chronic in ~75% of all patients[1], but direct acting antivirals (DAAs) achieve virus elimination in >95% of cases, rendering chronic hepatitis C a curable disease (reviewed in ref. 2). Nevertheless, a protective vaccine remains elusive thus HCV still is a major public health concern with 50 million patients being infected and 240,000 dying from the disease in 2022[3]. Chronic HCV infections can last for decades but eventually end-stage liver disease or development of hepatocellular carcinoma (HCC) can create a need for liver transplantation (LTX). A severe complication post LTX

in the context of HCV is the development of fibrosing cholestatic hepatitis (FCH) which is marked by high viral titers and rapid fibrosis progression accompanied by only mild inflammation[4], occurring in 2-10% of LTX patients[5,6]. DAA treatment efficiency is also high in FCH patients[7,8]. HCV is a remarkably heterogenous virus with 8 genotypes (gt) and over 100 subtypes differing by up to 30% on a nucleotide sequence level[9]. This is enabled by the error prone genome replication of the virus resulting in a heterogenous group of viruses inhabiting each patient, the so called quasispecies[10]. This population structure

A full list of affiliations appears at the end of the paper. *A list of authors and their affiliations appears at the end of the paper.
✉e-mail: Volker.lohmann@med.uni-heidelberg.de

helps the virus to escape the host immune response, but beyond that, phenotypic characterisations of HCV quasispecies evolution are scarce, one study reported a selection for variants with improved cell entry competence after LTX[11].

On a molecular level, HCV has a positive sense RNA genome with untranslated regions (UTRs) flanking a single open reading frame. The resulting viral polyprotein is cleaved by host and viral proteases into 3 structural and 7 non-structural (NS) proteins. While the structural proteins core, E1 and E2 together with p7 and NS2 are crucial for infectious particle production, they are dispensable for genome replication, essentially involving NS3-NS5B and the UTRs (reviewed in ref. [12]). Subgenomic replicons (SGRs) encompassing the HCV replicase, were established as the first HCV cell culture model based on the gt1b isolate Con1[13]. All isolates used for cell culture systems are based on consensus sequences derived from single patients, accounting for the risk of deleterious mutations being present in many individual viral clones due to the error prone replication process[14]. However, besides the gt2a isolate JFH1[15], all wildtype (WT) isolates needed to acquire cell culture adaptive mutations to replicate in hepatoma cells commonly used in cell culture[16]. One hotspot for these adaptive mutations was found in NS5A[17], a phosphoprotein without enzymatic function. Nevertheless, NS5A is involved in a variety of processes including determining the response to antiviral interferon treatment by accumulation of mutations in a 40 amino acid region termed interferon sensitivity determining region (ISDR)[18,19]. HCV cell culture systems were improved by the discovery that ectopic expression of the cytosolic lipid transporter SEC14L2 allowed for replication of WT isolates in cultured cells[20]. In this setup, we previously characterised a post-transplant gt1b isolate from an FCH patient, designated GLT1, which showed remarkably high replication fitness (RF) in cell culture[21].

Here, we show that the high RF evolved in the GLT1 patient after LTX. This elevated RF was mediated by a region in NS5A including the ISDR where we identified accumulation of mutations as the sequence signature of high RF. Furthermore, we identified high RF as a general feature of HCV in FCH patients arguing for a direct connection between the increase in viral replication and the severe course of disease. Finally, we observed that sequence signatures of high RF are rare in immunocompetent patients but found more frequently in the context of profound immunosuppression, such as occurs at the early stages of transplantation, indicating a complex interplay between viral RF and the host immune response.

## Results

### HCV sequence evolution in a patient who received two liver transplantations

We recently characterised the GLT1 isolate which showed remarkably high RF in cell culture[21]. GLT1 was isolated from a patient who received two LTXs, with both transplanted livers developing FCH, and represented the consensus sequence of the viral population after LTX2. To study the viral sequence evolution over the course of the two LTXs (Fig. 1A), we performed next generation sequencing (NGS) on two overlapping amplicons spanning the entire coding sequence of the genome (Fig. 1B). The most drastic change in the consensus sequence dominating at each time point was seen after LTX1 (Fig. 1C) where the HCV population became much more homogenous and remained so until the patient's death (Fig. 1D). To gain a more comprehensive view on distinct viral subpopulations, we next analysed sequencing data from individually cloned viral genomes (Fig. 1E, left panel). In line with the NGS data, phylogenetic analysis revealed again a more heterogenous population structure pre LTX1 with at least two distinct HCV subspecies (Fig. 1E, S1A). Both the bulk sequencing data and the analysis of individual viral clones indicated LTX1 having a strong impact on the viral population while LTX2 did not have a substantial effect on the quasispecies.

To understand the phenotypic implications of the observed sequence evolution, we characterised changes in RF, cell entry and infectious particle production over the course of the two LTXs. Post LTX, we saw no selection for increased cell entry in experiments with HCV pseudoparticles (HCVpp), in contrast to a previous study which identified a selection for increased entry competence after LTX[11] (Fig. S1B). Since HCV isolates apart from JFH1 require massive adaptation to allow infectious particle production[22], including the GLT1 isolate[21], we employed chimeric genomes encoding the JFH1 replicase fused to the pre- and post-transplant structural proteins to assess changes in infectious particle production[23] (Fig. S1C). While the pre LTX1β chimera was devoid of particle production, both other pre LTX variants showed a higher efficiency than the two post LTX chimeras. Generally, the data need to be interpreted with care due to compatibility issues in combining gt1b structural proteins with a gt2a replicase[24,25]. Still, we found no indication for a selection towards increased assembly or release efficiency in the post LTX structural proteins. To assess RNA replication fitness, we constructed reporter SGRs[26] harbouring the consensus sequence of the viral replicase (NS3-NS5B) for each timepoint (Fig. 1F, upper panel), and for both subspecies we found in the pre LTX1 sample (Fig. 1E). The resulting constructs were transfected into hepatoma cells lacking innate immune competence but expressing the lipid transporter SEC14L2 which is needed for replication of HCV WT isolates in cell culture[20]. Compared to the gt1b gold standard isolate Con1[13], both subspecies pre LTX1 showed a similarly low RF (Fig. 1F, lower panel). Strikingly, all SGRs based on the post LTX time points showed more than 50-fold increase in RF arguing for a selection towards high RF in the context of LTX.

Increased RF after LTX argued for a prominent role of both the replication space provided by the new liver and the immunosuppression introduced with LTX1 in exerting evolutionary pressure on the HCV quasispecies. Thus, genome replication fitness is a so far overlooked major selection factor for HCV upon LTX.

### High replication fitness is mediated by a small region in NS5A

We next set out to identify mutations crucial for the increase in RF post LTX (Fig. 1F). To this end, we created chimeric Con1 SGRs (low replicator) where one protein of the replicase was replaced by its GLT1 (high replicator) counterpart (Fig. 2A, upper panel). Transfer of GLT1 NS5A into Con1 was sufficient to elevate the RF of Con1 by more than 100-fold (Fig. 2A, lower panel). Transfer of NS5A subdomains into Con1 identified Low Complexity Sequence 1 and Domain 2 (LCS1D2) of GLT1 NS5A as being sufficient to boost the RF of Con1 to the level of GLT1 (Fig. 2B). LCS1D2 from Con1 further converted the high replicator GLT1 into a low replicator. These results argued for LCS1D2 being the main determinant of HCV RF. Splitting up LCS1D2 as well as further mapping of the region underlined that mutations all over LCS1D2 except the C-terminus contributed to its effects on RF (Fig. S2A–C). Therefore, we defined the Replication Enhancing Domain (ReED) to encompass the whole LCS1D2 region besides a highly conserved patch at the N-terminus (Fig. 2C). Interestingly, the ReED contains the ISDR, an interaction site with the essential host factor cyclophilin A[27] and a disputed binding domain for protein kinase R[28,29] (Fig. S2B).

In the GLT1 patient, we could see a high number of amino acid differences in the ReED between the two subspecies dominating pre LTX1 and GLT1 (Fig. 2C). To functionally characterize ReED evolution, we created chimeric SGRs of pre LTX1α and pre LTX1β containing the GLT1 ReED which dominated post LTX2. Indeed, the GLT1 ReED boosted replication of the two pre LTX subspecies 76-fold and 22-fold, respectively (Fig. 2D), confirming that selection for mutations in the ReED resulted in the high RF post LTX in the GLT1 patient.

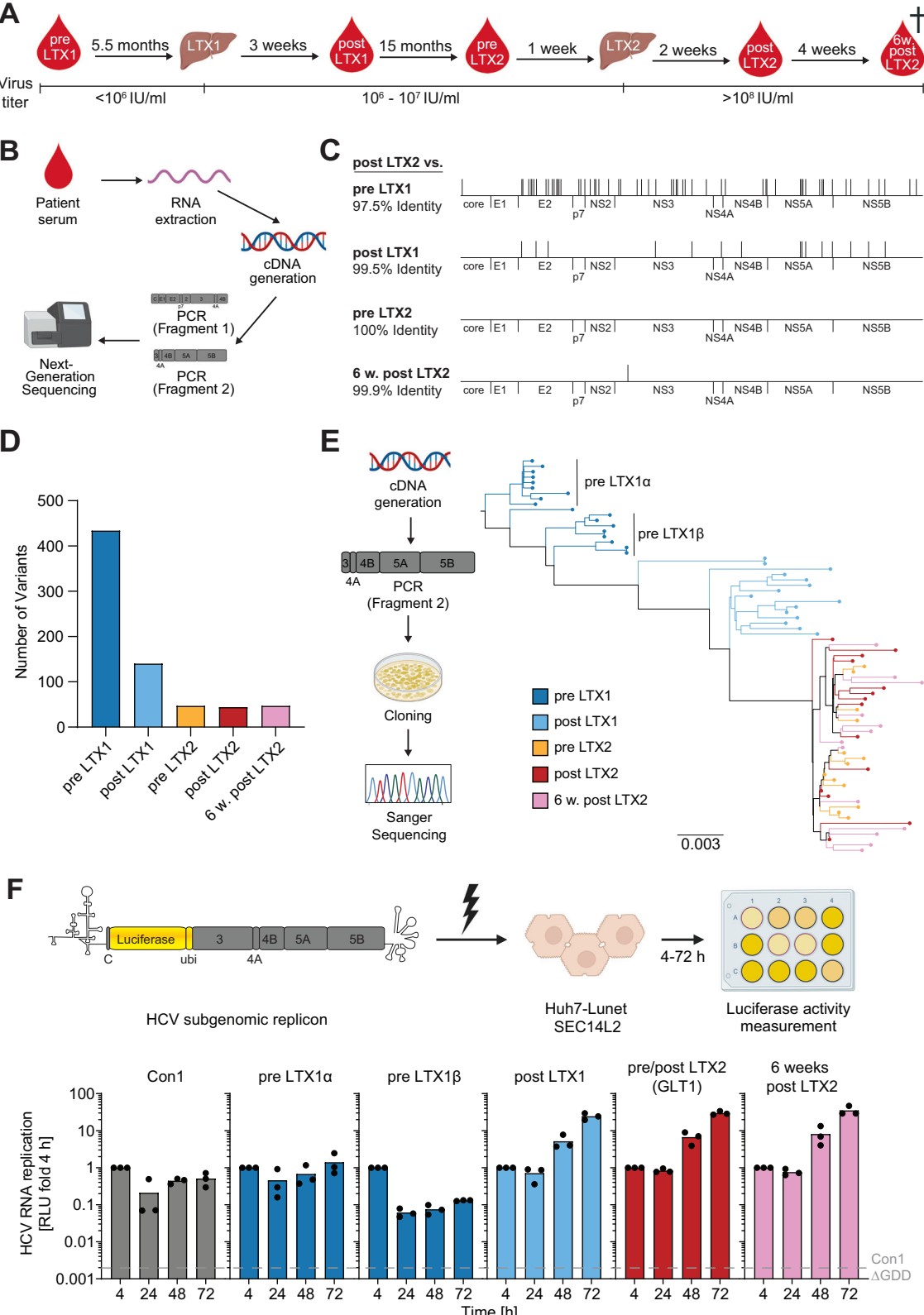

## Fibrosing cholestatic hepatitis is associated with highly replication fit HCV

To assess whether increased RF was a general feature of viral isolates evolving after LTX, we focused on a cohort of 22 additional HCV gt1b infected LTX patients of which 13 developed FCH post LTX[30,31]. Sequence analysis revealed a significantly higher divergence from the gt1b consensus for the ReED of FCH patients than for non-FCH patients, which was only driven by accumulation of mutations in the ISDR but not in the C-terminal part (C-term) of the ReED (Fig. 3A). All FCH patients had 3 or more ISDR mutations while the non-FCH patients had a maximum of 1 ISDR mutation rendering accumulation of ISDR mutations a clear sequence signature of FCH (Fig. 3A; S3A). To assess the RF of the FCH and non-FCH patients in cell culture, we created chimeric Con1 SGRs encoding the ReEDs of the post LTX

**Fig. 1 | HCV evolution in the GLT1 patient. A** Schematic illustrating time points of collection and viral load of longitudinal serum samples from the GLT1 patient. Note that the patient passed away 6 weeks post LTX2. **B** Schematic illustrating the workflow to retrieve bulk sequence information about HCV from patient sera. We extracted viral RNA from the serum, converted it into cDNA, amplified the HCV genome via PCR and performed next generation sequencing (NGS). For technical reasons, the HCV genome was amplified in two fragments spanning the entire coding sequence of the viral genome. **C** Amino acid alignments of the consensus sequences at the indicated time points with the post LTX2 ( = GLT1) sequence, differences are indicated by black dashes. **D** Number of DNA variants (QD > 1) in the HCV population of each time point. **E** Schematic of generation of sequencing data on the level of individual viral clones (left panel) and resulting phylogenetic tree for the second PCR fragment encompassing most of the viral replicase (right panel). **F** SGRs of the indicated constructs were electroporated into Huh7-Lunet SEC14L2 cells (upper panel), luciferase activity in cell lysates (RLU) was quantified as a correlate of RNA replication efficiency at the given time points and normalised to 4 h to account for differences in transfection efficiency. Con1 ΔGDD served as a replication deficient negative control. Data are from three independent biological replicates measured in technical duplicates. Each dot depicts the result of one replicate and the bar indicates the mean of all replicates. Schematics in **A**, **B**, **E**, **F** were created in BioRender. Lohmann, V. (2025) https://BioRender.com/upia2pp.

patients (Fig. 3B, upper panel). A chimera harbouring the gt1b consensus ReED, based on 358 gt1b full-length sequences, showed low RF, comparable to Con1 (Fig. S3B), indicating that the majority of hepatitis C patients is infected with a low replicator HCV. Strikingly, constructs harbouring the ReED from FCH patients replicated on average more than 100-fold higher than the non-FCH patient derived constructs, with only two exceptions (Fig. 3B, S3B). We verified that chimeric Con1 SGRs with patient-derived ReEDs accurately represented the RF of a patient's HCV isolate, by comparing the RF of four chimeras with SGRs encoding the patient's complete replicase (Fig. 3C) and confirmed the ReED's effect on RF in full-length HCV genomes (Fig. S4A). Overall, our data argued for a connection of accumulation of mutations in the ISDR with high RF in cell culture and the highly pathogenic course of liver disease after LTX.

For HCV gt1b isolates with available complete replicase sequences[21,30,31], phylogenetic analysis showed FCH-associated variants at various positions across the tree, indicating that probably any isolate might be able to evolve into a high replicator variant by acquiring ReED mutations (Fig. S4B). The genetic diversity determined through NGS of an NS5B fragment[30] decreased in almost all cases upon LTX, irrespective of disease outcome (Fig. S5A). However, observations similar to GLT1 were noted in only 3 patients, where a low replicator variant evolved into a high replicator variant after LTX. In contrast, 7 FCH patients already exhibited a high replicator sequence before undergoing LTX (Fig. S5C). Interestingly, torque teno virus (TTV) load in post LTX sera as a surrogate marker for immunocompetence[32,33], revealed no differences between non-FCH and FCH patients (Fig. S5B). This highlights the complexity of HCV evolution but overall underlines that acquisition of a high replicator HCV variant in late chronic infection or upon LTX was decisive for FCH development.

Since all currently approved DAA therapy regimens include NS5A inhibitors, we next assessed whether a high or low replicator ReED would impact on their inhibitory capacity. To this end, we compared the dose response for currently approved NS5A inhibitors on Con1 and GLT1 SGRs, as well as on a Con1 chimera with the GLT1 ReED and a GLT1 chimera harbouring the Con1 ReED. For treatment with the NS5A inhibitors Pibrentasvir, Velpatasvir and Daclatasvir, a surprising enhancement of RF for low replicators at very low drug doses was observed, with a peak at 1 pM for Pibrentasvir and Velpatasvir and 10 pM for Daclatasvir, respectively, somehow phenocopying the replication enhancing effect of the GLT1 ReED (Fig. 3D, S6A). This precluded assessment of distinct inhibitory concentrations (IC) expressed in IC50 or IC90 values. However, almost full inhibition of all variants was achieved already at 100 pM, suggesting that differences in RF by ReED variants will not affect the efficiency of currently used NS5A inhibitors. In line with this data, treatment with the NS5B inhibitor Sofosbuvir revealed no RF dependent differences in IC50 values (Fig. S6B). These results as well as the efficient response of FCH patients to DAA therapy[7,8] argue against an impact of RF on DAA treatment. Along these lines, we identified a post LTX isolate with a high replicator ReED in a patient who was successfully treated with Daclatasvir, Sofosbuvir and Ribavirin three months after LTX (Fig. S6C), thereby potentially preventing the establishment of FCH and showing that high replicator phenotypes will likely not interfere with treatment success.

All patients of the FCH cohort showed very high serum titers (>10,000,000 IU/ml, Fig. S5D), a hallmark of FCH[4], which we wanted to confirm on the level of HCV antigen load in the liver. We therefore used immunohistochemistry to compare the number of HCV positive cells in the liver of FCH and non-FCH patients. Indeed, HCV positive cells were significantly more abundant in liver samples of FCH patients, arguing for high intrahepatic viral replication (Fig. 3E, S7) which is in line with previous studies[34,35]. Interestingly, FCH4, containing three ISDR mutations post LTX (Fig. S3A), but having a low replicator phenotype in cell culture (Fig. 3C), showed exceptionally high antigen loads (Fig. S7B), indicative of a high replicator variant in vivo. This suggested that our assessment of RF in cell culture widely reflected replication competence in vivo but might have missed some high replicator variants. This data furthermore indicated that liver pathogenesis was directly or indirectly caused by high intrahepatic virus replication.

Overall, we could show a strong correlation between accumulation of ISDR mutations leading to increased viral RF and a severe course of disease for LTX patients.

## The ISDR and especially its first residue are key drivers of elevated replication fitness

The FCH cohort revealed a strong connection between ISDR mutations and elevated RF (Fig. 3A) while chimeric SGRs between Con1 and GLT1 indicated that the ISDR on its own cannot completely transfer the high RF of GLT1 to Con1 (Fig. S2C) arguing for an important role of the ReED C-term. However, the GLT1 C-term had 6 deviations from the gt1b consensus while other high replicator ReEDs like those from patients FCH2 or FCH3 were much closer to the consensus only harbouring 2 or 3 C-term mutations, respectively (Fig. S3A). To further assess the importance of the ReED C-term, we generated replicons harbouring the gt1b consensus C-term combined with the variable ISDR regions of FCH patients. In case of FCH2 and FCH3, the constructs only containing the ISDR replicated at the same level as the SGRs with the complete FCH2/FCH3 ReEDs, indicating that the ISDR mutations are indeed the main drivers of increased replication fitness (Fig. 4A). For GLT1, still a lower RF was observed when only the ISDR and not the complete ReED was present, underpinning the concept that the entire ReED of an isolate needs to be included in phenotypic analyses, since mutations in the C-term can contribute to its function. Next, we introduced a representative set of ISDR point mutations exclusively found in high replicator ReEDs into the Con1 SGR with the gt1b consensus ReED, revealing that P2209L (the first residue of the ISDR) but none of the other tested mutations could strongly enhance RF (Fig. 4B). Also, all other mutations found at this position in high replicator ReEDs presented a hydrophobic amino acid (P2209A/V/I) and enhanced RF to a various extent, but less efficiently than P2209L (Fig. 4C). Reverting residue 2209 back to the wildtype proline in several high replicator ReEDs drastically reduced RF further underlining the importance of residue 2209 (Fig. 4D).

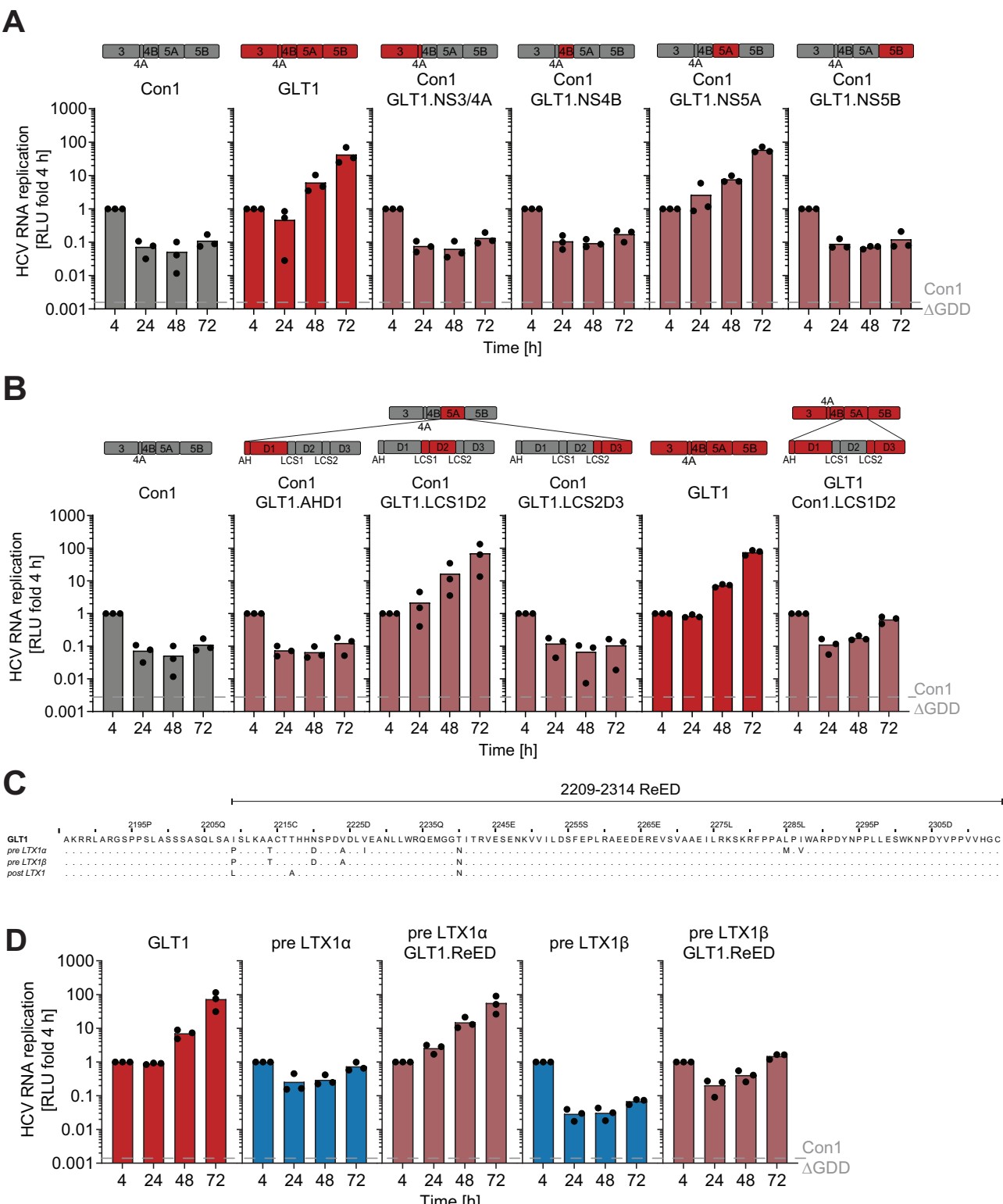

**Fig. 2 | Viral determinants of high replication fitness. A, B, D** SGRs of the indicated constructs were electroporated into Huh7-Lunet SEC14L2 cells, luciferase activity in cell lysates (RLU) was quantified as a correlate of RNA replication efficiency at the given time points and normalised to 4 h to account for differences in transfection efficiency. Con1 ΔGDD served as a replication deficient negative control. Data are from three independent biological replicates measured in technical duplicates. Each dot depicts the result of one replicate, and the bar indicates the mean of all replicates. **C** Amino acid alignment of LCS1D2 between GLT1 and the respective subspecies pre LTX1, dots indicate an amino acid being identical to GLT1.

One striking feature of some high replicator ISDRs were amino acid insertions as observed in FCH1, 5 and 11 (Fig. S3A). Intriguingly, FCH1&11 shared the same 4 amino acid insertion which, when introduced into a Con1 SGR with the low replicator gt1b consensus ReED, enhanced RF by 10-fold, arguing for an important role of this insertion in high RF. This data further demonstrated that an insertion can act in combination with additional point mutations (Fig. 4E). The insertion found in FCH5 was already

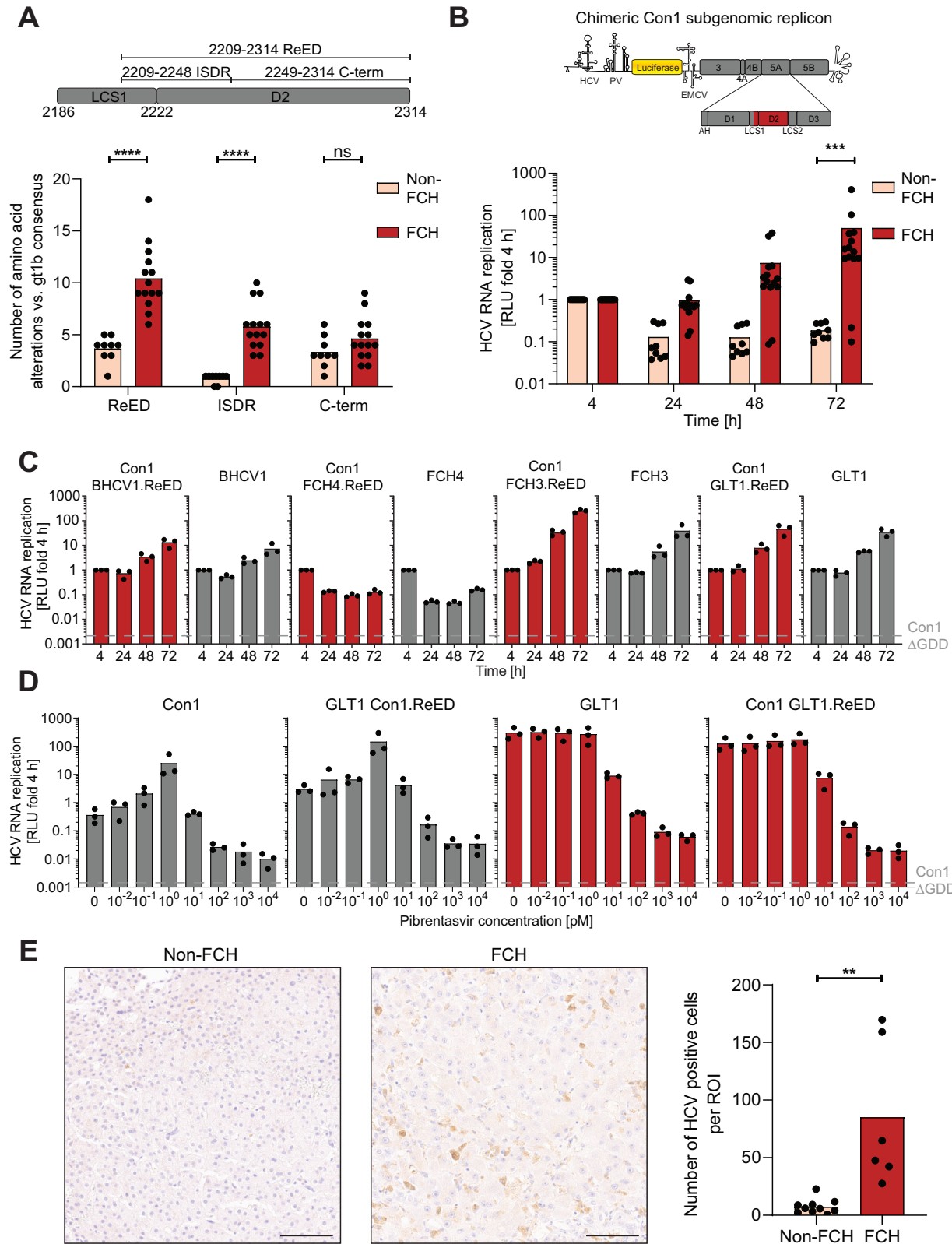

present prior to LTX and much larger (16 amino acids), but its introduction into Con1 with the gt1b consensus ReED proved to be detrimental for replication. Interestingly, the FCH5 insertion matched well to a fragment of the gt1b consensus sequence of the E1 protein, implying a recombination event during HCV replication.

Overall, the results highlighted the importance of the ISDR as the main driver of high RF with rather auxiliary functions of the ReED C-term. Only mutations at position P2209 and a small insertion were capable of substantially enhancing replication as single mutations, demonstrating that the increase in RF by the ISDR indeed mostly relies on accumulating effects of several mutations.

**Fig. 3 | Genetic and phenotypic analysis of the FCH cohort. A** Amino acid deviations from the gt1b consensus sequence were counted for each patient sequence post LTX, an insertion was counted as one mutation, irrespective of its length. The analysed regions are depicted in a schematic (upper panel). Each dot depicts the result of one patient, and the bar indicates the mean of all patients. 14 FCH patients were compared to 9 non-FCH patients. **B** Con1 based chimeric SGRs harbouring the ReED of a patient (upper panel) were electroporated into Huh7-Lunet SEC14L2 cells, luciferase activity in cell lysates (RLU) was quantified as a correlate of RNA replication efficiency at the given time point and normalised to 4 h to account for differences in transfection efficiency. Data are from three independent biological replicates measured in technical duplicates. Each dot represents the mean RLU of one patient and the bar indicates the mean of all non-FCH/FCH patients. 14 FCH patients were compared to 9 non-FCH patients. **A, B** Statistical significance was determined with a two-sided Mann–Whitney-U-test. ns = not significant, *** = p < 0.001, **** = p < 0.0001. Exact p-values: p = 0.00004 (**A**, ReED), p = 0.00001 (**A**, ISDR), p = 0.116 (**A**, C-term) and p = 0.0005 (**B**). **C–D** SGRs of the indicated constructs were electroporated into Huh7-Lunet SEC14L2 cells, luciferase

activity in cell lysates (RLU) was quantified as a correlate of RNA replication efficiency at the given time points and normalised to 4 h to account for differences in transfection efficiency. Con1 ΔGDD served as a replication deficient negative control. Data are from three independent biological replicates measured in technical duplicates. Each dot depicts the result of one replicate and the bar indicates the mean of all replicates. **D** Replication 96 h after electroporation is depicted, treatment with the indicated concentration of Pibrentasvir was performed 24 h after electroporation. **E** Sections of human livers from post-transplant patients either developing FCH or not were stained via IHC with an HCV NS5A targeting antibody. Representative image from one non-FCH (left panel, patient non-FCH12) and one FCH patient (middle panel, patient FCH14), scale bar represents 100 μm. Quantification of HCV positive cells per 500 × 500 μm region of interest (ROI) (right panel). Six regions per patient were analysed. Each dot depicts the mean result of one patient and the bar indicates the mean of all non-FCH/FCH patients. 6 FCH patients were compared to 10 non-FCH patients. Statistical significance was determined with a two-sided Student's t-test. ** = p < 0.01. Exact p-value: p = 0.0014.

## The ReED is a regulator of replication fitness in all major genotypes

Our study so far was focused on HCV gt1b. However, HCV is a very heterogenous virus with 8 gts and over 100 subtypes[9]. To address the impact of the ReED on RF of other major genotypes, we retrieved ReED sequences from LTX patients infected with gt1a, gt3a or gt4a from the HCV Research UK cohort[36] as well as two post LTX samples from gt1a infected patients developing FCH[30,37]. Most sequences had at least 3 ISDR mutations when compared to the gt specific consensus sequence. To study the impact of these ReEDs on RF, we used established isolates derived from chimpanzee infectious clones as SGR backbones: H77 (gt1a)[38,39], S52 (gt3a) and ED43 (gt4a)[40]. Patient derived ReED sequences as well as the gt specific consensus sequence were introduced into the respective isolates. RF of all constructs with a consensus ReED was low but was increased for ReEDs with an altered ISDR in many cases (Fig. 5A–C, S8-S10). The increases in RF were moderate compared to gt1b due to the cell culture systems being less suitable for non-gt1b isolates[41] (reviewed in ref. [14]). Still, for gt1a, 9 out of 11 isolates with altered ISDR replicated at least 2-fold higher than the consensus (Fig. 5A, S8B), while in gt3a 11 out of 17 (Fig. 5C, S9B) and in gt4a 2 out of 6 isolates showed this phenotype (Fig. S10). H77 based SGRs harbouring the ReED of the gt1b high replicator GLT1 showed a 20-fold higher RF than H77 with the Con1 ReED (Fig. S8C), highlighting that the high replicator property of a ReED is conserved between subtypes. For gt1a and gt3a, we also tested the impact of common mutations found in high replicators of the respective genotypes. Similar to gt1b, P2209L emerged as the point mutation inducing the strongest increase in RF (Fig. 5B, D, S8D, S9C). Overall, our experiments revealed that the ReED can regulate the replication fitness of all major HCV genotypes.

## Frequency of potential high replicators in different clinical contexts

So far, by mainly focusing on LTX patients we could show that the ReED can regulate RF in all major gts with accumulation of ISDR mutations being the sequence signature of high RF. This sequence signature allowed us to investigate patients of the HCV Research UK cohort containing HCV sequence information and detailed clinical annotations for more than 2000 patients, mainly infected with the major HCV gts 1a, 1b and 3a[36]. For these gts, 12–15% of patients harboured a potential high replicator defined by having 3 or more amino acid mutations in the ISDR when compared to the respective gt specific consensus sequence (Fig. 6A). Because of the similar prevalence of potential high replicators, we decided to perform all following analyses with a combined dataset of gt 1a, 1b and 3a patients. In line with our previous results, we found a significant enrichment of potential high replicators in patients sequenced after LTX (Fig. 6B). Mutations at residue 2209 which we showed to have a replication enhancing effect

presented a similar accumulation in post LTX patients (Fig. S11A). The enrichment of mutations was also specific to the ISDR, with no significant accumulation of mutations in the whole replicase (Fig. 6C). Early acute infections should create similar conditions as LTX, offering a naïve liver in absence of adaptive immune pressure. This might create a selective advantage for highly replicating variants but also trigger adaptive immunity more efficiently and thereby impact on infection outcome. Indeed, a higher proportion of potential high replicators was observed in patients who cleared the virus compared to patients who developed a chronic infection (Fig. 6D). Since samples from this disease stage are scarce due to the inapparent course of disease, no statistical significance was reached. Still, this data indicated that high RF might only be advantageous for HCV variants under conditions of immunosuppression, such as LTX, but rather facilitate clearance in immunocompetent individuals.

HCV serum titers of more than 10,000,000 IU/ml were common in FCH patients (Fig. S5D), suggesting a correlation with RF in immunosuppressed patients. Surprisingly, in non-transplant patients from the HCV Research UK cohort, there was no trend towards enrichment of potential high replicators in patients with titers exceeding 10,000,000 IU/ml (Fig. 6E) and no significant overall correlation between titer and ISDR mutations was found (Fig. 6F). This lack of correlation was confirmed in a cohort of pregnant women[42] (Fig. S11B), suggesting that HCV serum titers are not primarily determined by RF in immunocompetent patients.

In FCH patients, we saw a strong correlation between high RF and pathogenesis. But in non-LTX patients of the HCV Research UK cohort, no associations between the presence of potential high replicators and the liver status at the time point of sequencing, liver stiffness or past diagnosis of cirrhosis or decompensation were observed (Fig. S11C-E), as for other parameters like drug consumption, bilirubin levels, biological age or time since first infection (Fig. S11E, S12A–B). In contrast, HCV-HIV coinfection was a parameter significantly associated with increased presence of potential high replicators (Fig. 6G), which might again be linked to immunosuppressive conditions. Finally, in the subgroup of patients who ever had an HCC diagnosis in their life (Fig. S11E), we found that potential high replicators were significantly more common in patients after they received their HCC diagnosis (Fig. 6H). Thus, previous HCC diagnosis or HCV-HIV coinfection appear to be scenarios favouring the emergence of potential HCV high replicators.

To understand why only some patients harboured a high replicator, we turned towards host genetic determinants. In the context of the immune response to HCV, it was discovered that single nucleotide polymorphisms (SNPs) in the *IFNL4* gene coding for interferon lambda 4 (IFNλ4) are associated with disease outcome[43]. To this end, we analysed a cohort of 1481 patients with available host genetic and virus

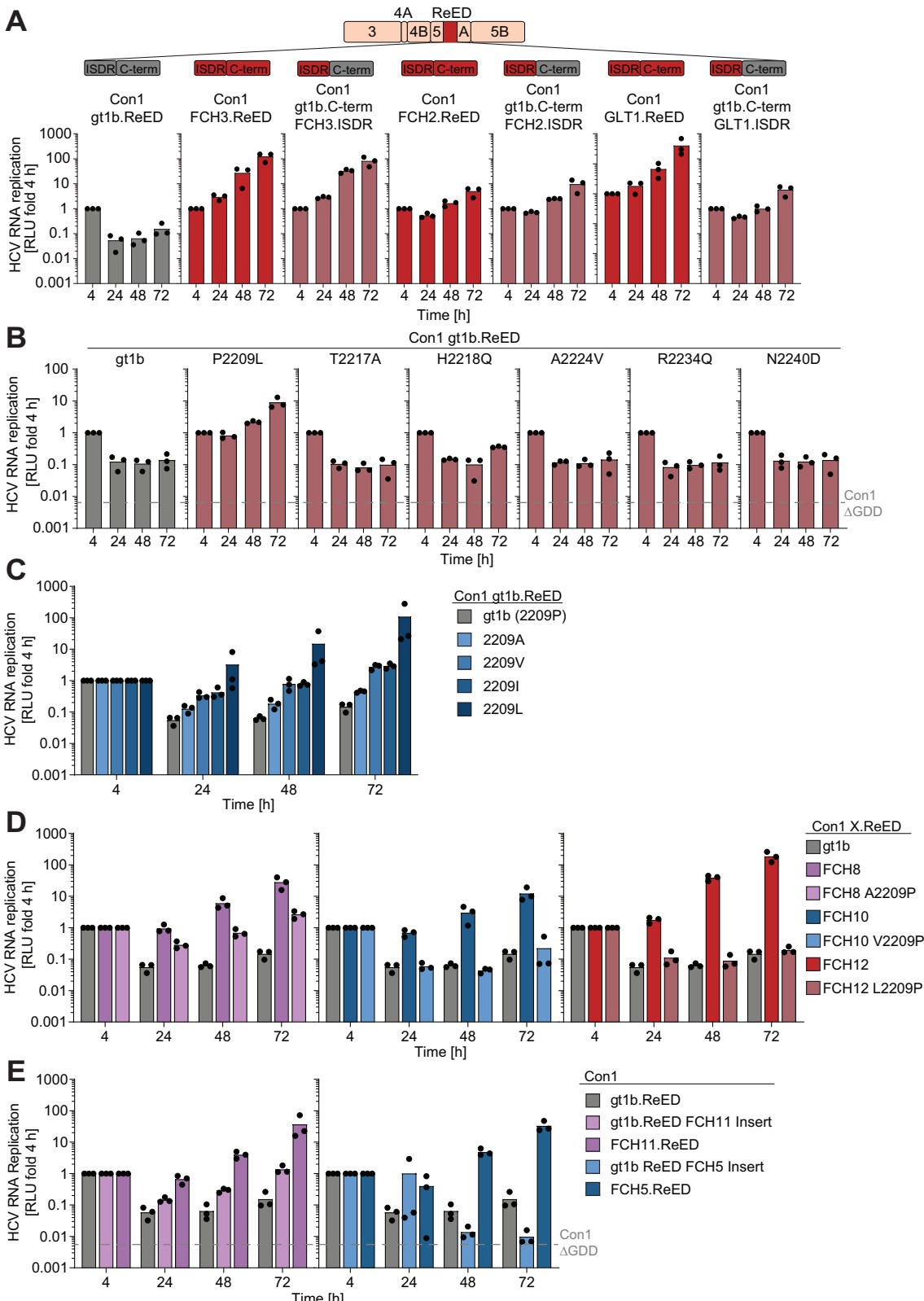

sequence information[36,44,45]. Here, we found that potential high replicators were more prevalent in patients with the CC genotype for *IFNL4* SNP rs12979860, favouring spontaneous clearance (Fig. 6I). Beyond *IFNL4*, a genome wide association study did not reveal any other host genetic determinants significantly associated with the presence of potential high replicators (Fig. S12C). Thus, the importance of SNPs in

the *IFNL4* gene further highlight the tight interplay of the immune response and RF.

Overall, our data highlight that the emergence of potential high replicators can correlate also with clinical parameters beyond the LTX context like HIV coinfection or HCC diagnosis as well as the genetic background of the *IFNL4* gene.

**Fig. 4 | Detailed sequence determinants of elevated replication fitness. A** Con1 based SGRs harbouring either patient derived ReEDs or chimeric ReEDs combining a patient derived ISDR with the gt1b consensus C-term. **B**, **C** Chimeric Con1 SGRs containing the gt1b consensus ReED and the indicated point mutations in the ISDR. **D** Chimeric Con1 SGRs containing the indicated patient derived ReED or a variant of that ReED with residue 2209 mutated back to the WT proline. **E** Chimeric Con1 SGRs containing the gt1b consensus ReED and the indicated insertion in the ISDR. The ReEDs from which the respective insertions were derived served as reference.

**A–E** SGRs of the indicated constructs were electroporated into Huh7-Lunet SEC14L2 cells, luciferase activity in cell lysates (RLU) was quantified as a correlate of RNA replication efficiency at the given time points and normalised to 4 h to account for differences in transfection efficiency. Con1 ΔGDD served as a replication deficient negative control. Data are from three independent biological replicates measured in technical duplicates. Each dot depicts the result of one replicate, and the bar indicates the mean of all replicates.

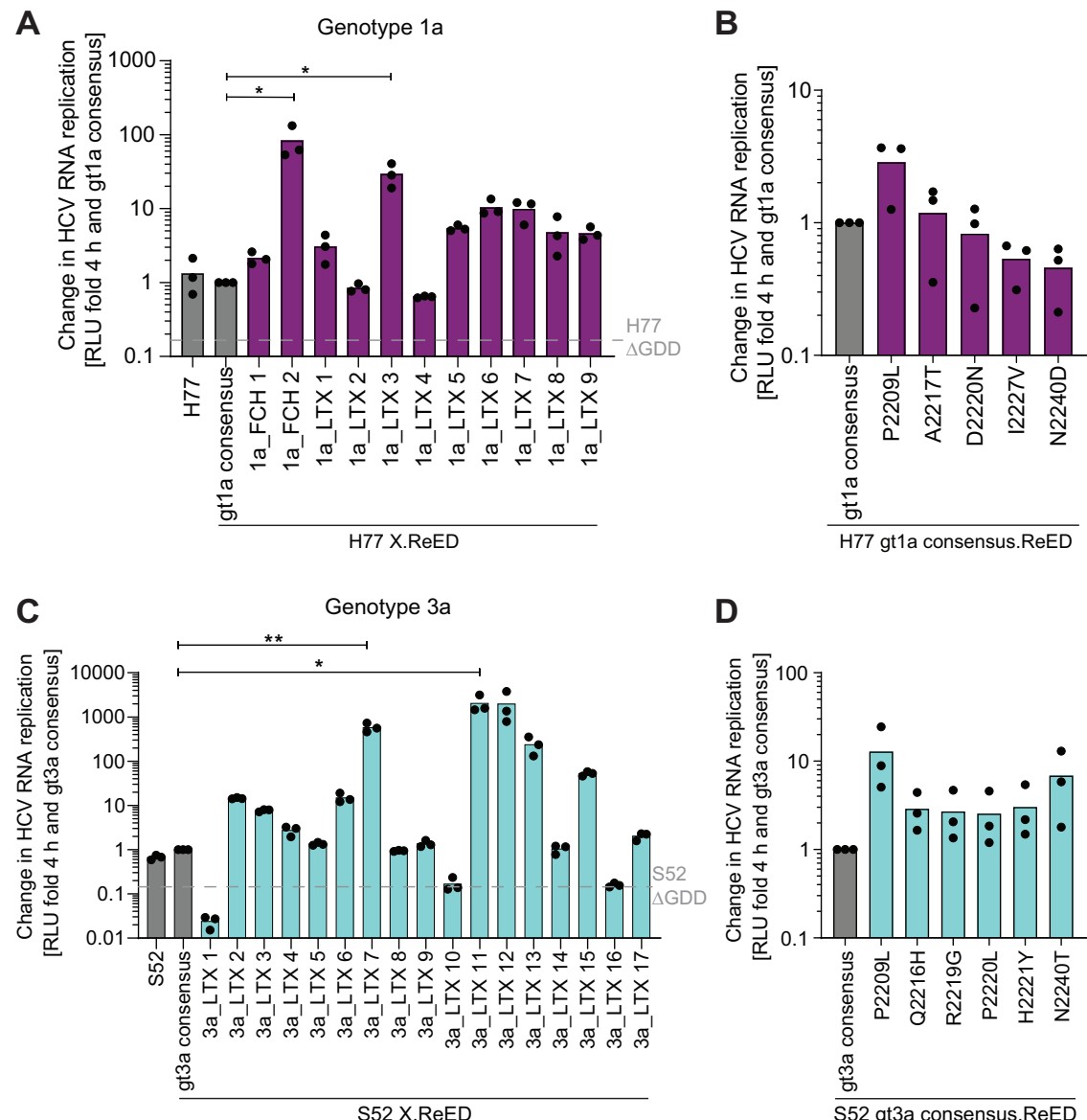

**Fig. 5 | The ReED can regulate replication fitness in all major genotypes.** H77 (**A**) or S52 (**C**) SGRs harbouring patient derived ReEDs either from FCH patients[30,37] or from LTX patients from the HCV Research UK cohort[36]. **B**, **D** Chimeric H77 SGRs containing the gt1a consensus ReED and the indicated point mutations in the ISDR (**B**) or S52 SGRs containing the gt3a consensus ReED and the indicated point mutations in the ISDR (**D**) were analysed. **A–D** SGRs of the indicated constructs were electroporated into Huh7-Lunet SEC14L2 cells, luciferase activity in cell lysates (RLU) was quantified as a correlate of RNA replication efficiency at the given time points and normalised to 4 h to account for differences in transfection efficiency

and subsequently normalised to the values for the gt specific consensus of the respective replicate. H77 ΔGDD (**A**) or S52 ΔGDD (**C**) served as a replication deficient negative control. Data are from three independent biological replicates measured in technical duplicates. Each dot depicts the result of one replicate, and the bar indicates the mean of all replicates. Statistical significance was determined with a two-sided Student's t-test. ns = not significant, * = p < 0.05, ** = p < 0.01. Exact p-values: p = 0.03 (A, FCH1a_2), p = 0.01 (A, 1a_LTX_3), p = 0.001 (C, 3a_LTX7), p = 0.02 (C, 3a_LTX11).

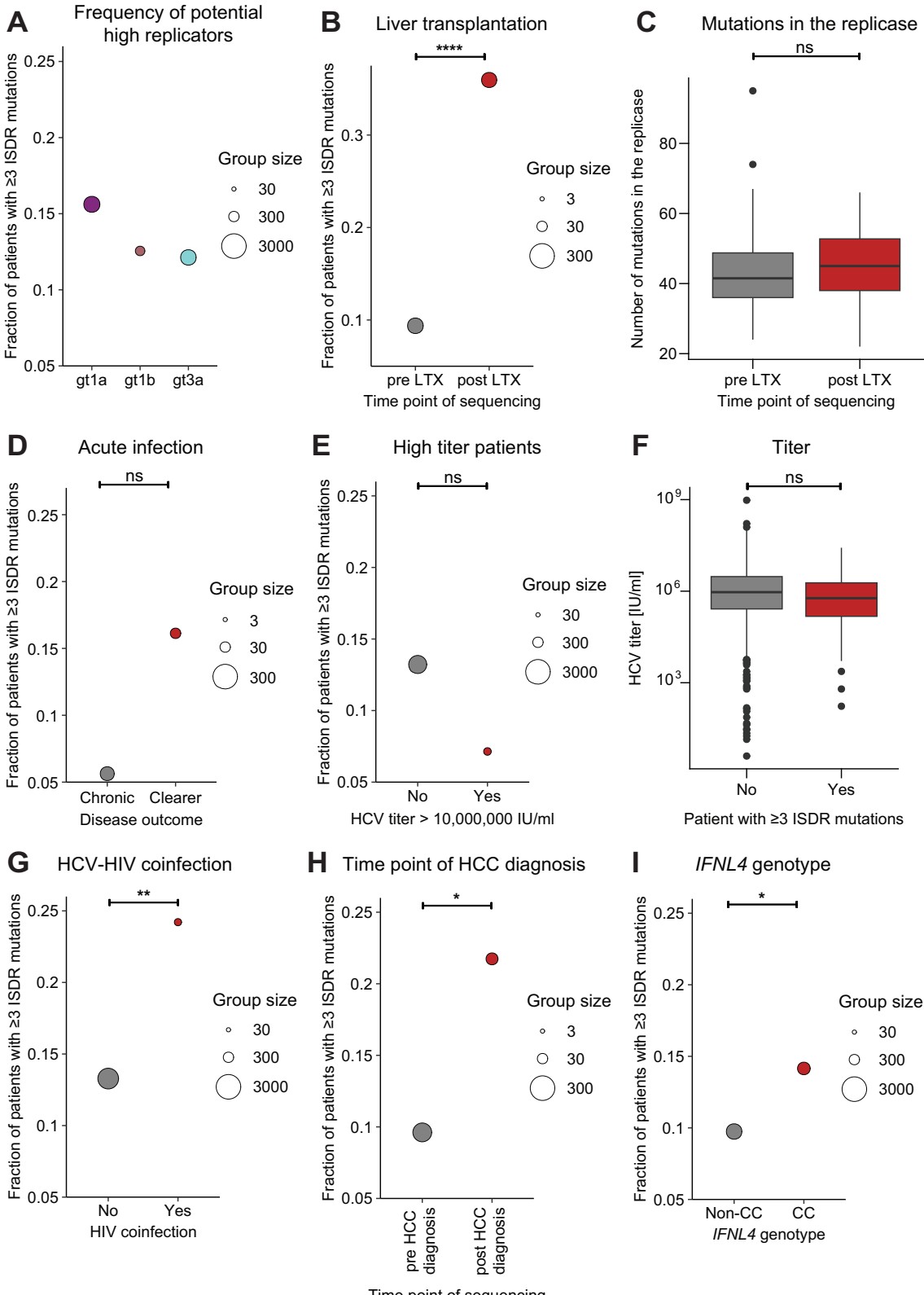

## Discussion

In the present study, we identified a region within NS5A, termed ReED, conferring high genome replication fitness to clinical isolates of the main HCV genotypes. Accumulation of mutations in a portion of the ReED, known as ISDR, was the sequence signature of high replicator variants. HCV variants with high replicator signatures were enriched under conditions of immune suppression and associated with severe pathogenesis after LTX.

HCV pathogenesis is usually considered to be immune mediated[46]. It was previously speculated that direct cytopathic effects by HCV could be causative of FCH[47,48], a hypothesis that is strongly supported by our data, showing 100-fold higher genome replication

**Fig. 6 | Sequence signatures of high RF in various clinical contexts.** For each patient's ISDR sequence, the number of amino acid differences compared to the gt specific consensus sequence was determined. If an ISDR had 3 or more mutations, the patient was considered to be infected with a potential high replicator. **A–C, E–H** Data based on the HCV Research UK cohort[36]. **A** Fraction of potential high replicators stratified by gt. **B–I** Data for gts 1a, 1b and 3a was pooled, for all analyses the results for the individual gt had the same trend as the combined analysis. **B** Fraction of potential high replicators in patients who ever received an LTX depending on whether samples for HCV sequencing were collected pre or post LTX. **C** In the patients depicted in panel **B**, number of mutations in the replicase (excluding the ReED) compared to the gt specific consensus. **B, C** Comparison of 94 pre LTX and 86 post LTX patients. **D–I** Post LTX patients were excluded from the analysis to prevent confounding the results. **D** Sequences from the HITS-p cohort[76] combined with 28 patients participating in the Montreal Hepatitis C cohort (HEPCO)[96] sequenced for this study, informing about the fraction of potential high replicators in acute patients who either developed a chronic HCV infection or cleared the virus without antiviral treatment. Comparison of 71 patients developing a chronic infection and 31 clearing the infection. **E** Relative abundance of potential high replicators in the context of very high serum titers (>10,000,000 IU/ml).

Comparison of 1407 low titer and 112 high titer patients. **F** Correlation of the presence of potential high replicators and HCV serum titers. 193 potential high replicators were compared to 1295 potential low replicators. The most recent titer measurement was used and only patients with an available titer measurement within a year of sequencing were included. **G** Fraction of high replicators in patients with an HCV-HIV coinfection. Comparison of 1980 patients without coinfection and 95 patients with coinfection. **H** Fraction of high replicators in patients who ever had an HCC diagnosis depending on whether samples for HCV sequencing were collected pre or post HCC diagnosis. Comparison of 156 patients sequenced before HCC diagnosis and 46 patients sequenced after HCC diagnosis. **I** Combined data from patients of the HCV Research UK[36], STOPHCV[45] and BOSON[44] cohorts where host genomic sequence information was available, showing the fraction of potential high replicators depending on the host's *IFNL4* rs12979860 genotype. Comparison of 944 non-CC and 537 CC patients. **C, F** The centre line signifies the median, the box the 25th and 75th percentile, and the whiskers the 1.5 interquartile range. Statistical significance was determined using a two-sided Student's t-test (**C, F**) or a Fisher's exact test (**B, D, E, G–I**). ns = not significant, * = p < 0.05, ** = p < 0.01, **** = p < 0.0001. Exact p-values: p = 1.478e-05 (**B**), p = 0.005 (**G**), p = 0.04 (**H**), p = 0.01 (**I**).

---

fitness of isolates dominating in FCH patients. To date, only increased cell entry competence was identified after LTX[11], but our study demonstrates that high genome replication fitness can be another so far unrecognised selection factor, decisive for disease outcome. Interestingly, the presence of mutations enhancing viral fitness were also described for hepatitis B virus (HBV) patients developing FCH[49,50]. Since HCV associated FCH currently mainly occurs in the context of HCV negative patients receiving an HCV positive organ[51], sequencing the viral quasispecies could allow an early identification of patients at risk.

On a sequence level, we were able to show that accumulation of mutations in the ISDR, which is the N-terminal part of the ReED, is a sequence signature of high RF. The ISDR was initially identified in the context of response to interferon (IFN) treatment in HCV gt1b infected patients in Japan[18]. The IFN sensitive "mutant type" was defined by at least 2 to 4 ISDR mutations[52–55] which is in line with our cutoff value of 3 or more ISDR mutations being associated with FCH and high RF. Previous studies already suggested increased replication for gt1b by ISDR mutations[56,57], but were limited by a small sample size and low magnitude of effects in the implemented cell culture models. Our study now highlights the general importance of the ReED in dramatically increasing RF of the major gts 1a, 1b, 3a and 4a, drawing on samples from a diverse range of geographical backgrounds.

For gt1b, the ISDR point mutations tested here were among the most common ISDR mutations identified in larger cohorts and they were shown to have a strong association with IFN treatment response[52,58]. Interestingly, P2209L and A2224V were previously observed to enhance RF in gt1b but again in a more limited cell culture setup[56]. In our study, mutations at residues 2209 and 2240 were commonly found in high replicators of gts 1a, 1b and 3a, arguing for a certain degree of conservation of ReED residues with functional relevance, even though only mutations at position 2209 showed a consistent replication enhancing effect. The conserved nature of the ReED is supported by the fact that a gt1b high replicator ReED was able to increase replication of gt1a H77. Another contributor to high RF identified here was a 4 amino acid insertion after residue 2214. This was in line with data on the HCV-N SGR where an insertion in the ISDR was characterized to enhance RF[57]. The ISDR was already shown to tolerate insertions and deletions in cell culture, although without replication enhancing effects[59]. Overall, the failure of most point mutations in the ISDR in inducing increases in RF highlights that cumulative effects of ISDR mutations appear to drive high RF.

While all dominant HCV variants in FCH patients had high replicator sequence signatures, FCH4&9 showed no high replicator phenotype in cell culture. Since FCH4 had the highest HCV antigen load in the liver and both isolates were associated with very high serum titers, it appears likely that they still represent high replicators in vivo. FCH4&9 had a relatively high genetic diversity compared to the whole FCH cohort, based on available NGS data from the NS5B coding region[30]. Thus, divergent high replicator subtypes in the quasispecies might have remained unrecognised in the available NS5A consensus sequence. It furthermore remains elusive how the ReED affects RF. Domain 2 of NS5A which encompasses the C-terminal 80% of the ReED, was shown to interact with the HCV RNA dependent RNA polymerase NS5B, modulating polymerase activity[60–62]. The variability of independent substitution and insertion patterns observed in high replicator variants overall renders a loss of function more likely than a gain of function. We therefore hypothesize that the consensus ReED is a negative regulator of NS5B activity with the mutations observed in high replicators unleashing a constitutively active NS5B. This hypothesis is in agreement with the fact that treatment with NS5A inhibitors at low picomolar concentrations increased replication of low replicator variants, phenocopying high replicator ReEDs. Low picomolar concentrations of NS5A inhibitors therefore might inactivate a subspecies of NS5A molecules, releasing their inhibitory activity on NS5B by a similar mechanism as the ReED. However, further studies are required to understand how the ReED governs HCV replication fitness on a molecular level.

New replication space provided by the transplanted liver combined with suppression of adaptive immunity are plausible determinants providing a selective advantage for high replicator variants. HCV associated FCH was also diagnosed in kidney[51], heart[63], lung[64] and hematopoietic stem cell[65] transplant and AIDS patients[66], arguing for immunosuppression as the main driver of evolution towards high RF. HCV-HIV coinfection is an additional condition with increased proportion of potential high replicator variants identified in our study, that might be linked to immunosuppression due to AIDS (reviewed in ref. 61). However, the HCV Research UK cohort only contains information on HIV coinfection but lacks further clinical details. Future studies therefore should envisage to identify HCV-HIV coinfected individuals in clinical cohorts allowing an in-depth characterization of immune status and the presence of HCV high replicator variants. Additionally, potential high replicators were more frequent in patients after receiving their HCC diagnosis which could be linked to immunosuppressive effects of cancer treatment, in line with a report of FCH upon chemotherapy[67], or the tumour microenvironment (reviewed in ref. 62). The impact of ISDR mutations on HCC development so far has been controversially discussed in literature, with one study observing an association of HCC with WT ISDR (<4 mutations)[68], whereas two previous reports showed a higher frequency of altered ISDRs in HCC

patients[55,69], in agreement with our data. To solve this discrepancy and to support our hypothesis on immunosuppressive anti-cancer therapies driving the development of high replicator variants in HCC, a cohort with more detailed clinical background on the patients will be required.

While weakened adaptive immunity might favour high replicator variants, it cannot fully explain the presence of high replicators in immunocompetent patients. Here, ISDR mutations increasing RF might compensate for the fitness costs of resistant variants upon DAA therapy or of immune escape variants emerging in chronic infection. The enrichment of HCV variants with high replicator signature in patients with the defective *IFNL4* CC allele we found in our cohort is further pointing to the complex interplay of viral replication fitness and host immunity. The CC allele was shown to favour cure upon IFN treatment and clearance of acute infections[43,70], since IFNλ4 hinders antigen presentation and the adaptive immune response[71]. We further found potential high replicators more frequently in acute patients who went on to clear the infection. HCV with high RF might be more vulnerable to the adaptive immunity due to higher antigen expression and therefore more likely being cleared in fully immunocompetent hosts. Along these lines, HCV clearance after waning of immune checkpoint inhibitors for cancer treatment[72] or after reduction of immunosuppressive therapy in two FCH patients[73,74] were reported elsewhere. Viral replication fitness thereby appears to be a key determinant of infection outcome. This needs to be considered for the choice of the infection inoculum used in controlled human infection models, currently planned to facilitate development of protective HCV vaccines[75].

In conclusion, our study shows that HCV RF is a selection factor particularly in immunocompromised patients. We identified a key regulator of RF within the HCV genome and could show a massively elevated RF in FCH patients, decisive for disease outcome. Our data thereby establish genome replication fitness as an important variable in the interplay between HCV and the immune system potentially contributing to clearance or persistence of the infection, with further potential implications for DAA treatment and vaccine development.

## Methods

### Patients
The HCV Research UK cohort[36], the cohort of pregnant women[42], the HITS-p cohort[76], the HEPCO[77], the GLT1 patient[21], the BHCV1 patient[31], patient 1a_FCH2[37], patients FCH1-12, non-FCH1-3 and 1a_FCH1[30] were described previously. Liver samples of patients non-FCH10-19 and FCH13&14 were provided by the tissue bank of the German Centre for Infection Research (DZIF, Heidelberg, Germany) and liver samples from patients FCH1, 4, 5 and 12 are from a previously described cohort[30]. Patients non-FCH4-9 were selected to be post LTX without FCH diagnosis and no successful antiviral treatment in the first year post LTX. Patients FCH1-12, non-FCH1-9, LTX_DAA1 and 1a_FCH1 are described in more detail in Table S2.

### Cell culture
Clones of the immortalised hepatoma cell line Huh7-Lunet either ectopically expressing CD81, SEC14L2 or an empty plasmid conferring blasticidin resistance were described previously[21]. They were cultured in Dulbecco's modified Eagle's medium (Gibco), supplemented with 1% (v/v) non-essential amino acids, 2 mM L-glutamine, 100 U/ml penicillin, 100 mg/ml streptomycin, 10% (v/v) heat-inactivated fetal-calf serum and 5 μg/ml blasticidin. All cell lines were regularly tested to check they were free of mycoplasma contamination using the MycoAlert Mycoplasma Detection kit (Lonza).

### Cloning
If not specified otherwise, DNA fragments for cloning were either generated via PCR using the PhusionFlash High-Fidelity Master Mix

(Thermo Fisher Scientific), via digest with appropriate restriction enzymes (New England Biolabs, Thermo Fisher Scientific) or via DNA synthesis (BioCat, Thermo Fisher Scientific). DNA fragments were combined either via T4 DNA ligase (Thermo Fisher Scientific) or with the NEBuilder HiFi DNA assembly cloning kit (New England Biolabs). Correct DNA sequence of the final plasmid was confirmed via Sanger or Oxford Nanopore sequencing (Microsynth AG). All plasmids used and generated during this study can be found in Table S3. The positioning of residues within the HCV polyprotein always refers to the gt1a H77 strain (GenBank accession AF009606).

### Sequencing of viral RNA extracted from patient sera
Viral RNA was purified from 200 μl patient serum using the NucleoSpin Virus kit (Macherey-Nagel). cDNA based on the extracted RNA was generated using the SuperScript IV First-Strand Synthesis kit (Invitrogen) with HCV genotype specific primers (Table S1). To amplify the DNA, nested PCR was performed using the SuperFi II DNA polymerase kit (Invitrogen) according to the manufacturer's instructions. A first 50 μl PCR reaction (primers see Table S1) with 25% of the cDNA as template was run for 30 cycles. 5 μl of this PCR product was used as template for the second PCR (primers see Table S1) running again for 30 cycles. The desired amplicon was purified via gel electrophoresis with the NucleoSpin Gel and PCR Clean-up kit (Macherey-Nagel). Fragments were sequenced either via Sanger sequencing (Microsynth AG) or NGS was performed using the MiSeq v2 (2x250bp) platform (Illumina) with an average sequencing depth of ~4000 reads (Microsynth AG).

### Analysis of NGS data
Sequence reads were aligned to the GLT1 WT reference sequence (GenBank accession OM222702) using the Burrows-Wheeler aligner (version 0.7.17)[78]. Variant calling was performed using the Haplotype-Caller of the genome analysis toolkit (version 4.1.4.1)[79]. Consensus sequences were derived with the consensus function of bcftools (version 1.22)[80].

### Phylogenetic analysis of individual viral clones
To retrieve individual viral clones from the GLT1 patient, the same amplicons used to derive the consensus sequences were cloned into a pCR-XL-2-TOPO vector using the TOPO™ XL-2 Complete PCR Cloning Kit (Invitrogen). Phylogenetic trees were constructed using the minimum evolution principle[81] with the tools of the virus and pathogen database and analysis resource (VipR)[82]. Trees were visualised using FigTree (version 1.4.4).

### In vitro transcription (IVT)
DNA plasmids were linearised with appropriate restriction enzymes cutting after the 3'UTR of the respective construct. DNA was purified using the NucleoSpin Gel and PCR Clean-up kit (Macherey-Nagel). IVT was performed by combining 10 μg linearised plasmid DNA, 6 μl T7 polymerase (homemade), 12.5 μl of rNTP solution (25 mM of each rNTP), 20 μl 5x RRL buffer (1 M HEPES, pH 7.5, 1 M MgCl$_2$, 1 M Spermidine and 1 M DTT), 100 U rRNasin RNase inhibitor (Promega) and filling up to 100 μl with water. The reaction was incubated at 37 °C overnight. 20 U RQ1 DNAse (Promega) was added the next day and incubated for 1 h at 37 °C. RNA was purified via phenol-chloroform extraction. Here, the reaction mixture was combined with 420 μl H$_2$O, 60 μl 2 M sodium acetate (pH = 4.5) and 400 μl phenol/water saturated (pH <5). After mixing and incubation on ice for 10 min, samples were centrifuged at 21,000 × $g$, 4 °C for 10 min. Supernatant was transferred into a new tube and mixed with one volume chloroform. The mixture was centrifuged for 10 min at 21,000 × $g$. Supernatant was again transferred into a new tube and mixed with 0.7 volumes isopropanol and subsequently centrifuged for 10 min at 21,000 × $g$. Supernatant was removed, the RNA washed

with 70% ethanol and resuspended in water. RNA quality was controlled via agarose gel electrophoresis.

## Electroporation

To transfect hepatoma cells with IVT RNA, electroporation was performed. After detaching the cells from the cell culture dish, they were washed once with PBS. Then, cells were resuspended in Cytomix (120 mM KCl, 0.15 mM CaCl₂, 10 mM K₂HPO₄/KH₂PO₄ (pH = 7.6), 2 mM EGTA, 25 mM HEPES, 5 mM MgCl₂ and freshly added 2 mM ATP, 5 mM glutathione) to reach a final concentration of $10^7$ cells/ml. Of this suspension, 200 µl ($2*10^6$ cells) were combined with 2.5 µg of the respective RNA to be transfected in an electroporation cuvette (gap width: 0.2 cm) and electroporated at 975 µF and 166 V using a Gene-Pulser system (BioRad). Electroporated cells were then resuspended in 6 ml DMEM. For the timepoints to be harvested 4 h and 24 h after electroporation, 1 ml of this mixture was put in a well of a twelve-well plate. For the 48 h and 72 h time points, 0.5 ml of the mixture and 0.5 ml of medium were combined in a well. Cells were then incubated at 37 °C and 5% CO₂.

## Drug treatment

For treatment with the antiviral drugs Sofosbuvir, Velpatasvir, Pibrentasvir (all MedChemExpress) and Daclatasvir (Bristol-Myers Squibb), electroporation was performed as described above. For seeding, 150 µl of resuspended electroporated cells were mixed with 350 µl medium in a twenty-four-well plate. After 24 h, cells were treated with the respective concentrations of antiviral drugs and harvested 96 h after electroporation.

## Luciferase assay

Samples were harvested by washing with PBS and subsequent lysis in 200 µl lysis buffer (1% Triton X-100, 25 mM glycyl glycine, 15 mM MgSO₄, 4 mM EGTA, and freshly added 1 mM DTT) per well. Firefly luciferase activity measurements were performed by mixing 80 µl lysate per technical replicate with 350 µl assay buffer (25 mM glycyl glycine, 15 mM K₃PO₄ buffer (pH = 7.8), 0.15 M MgSO₄, 4 mM EGTA (pH = 7.8) and freshly added 1 mM DTT, 2 mM ATP). During luminescence measurement, a Lubat LB9510 tube luminometer (Berthold Technologies) injected 100 µl of a luciferin solution (0.2 mM luciferin, 25 mM glycyl glycine) into the sample and then detected the signal for 20 s. Nano luciferase measurements were performed with the same device using the Nano-Glo luciferase assay system (Promega) according to the manufacturer's instructions.

## Particle production

To produce infectious particles, Huh7-Lunet CD81 cells were electroporated with IVT RNA as described above. One ml of the resulting suspension of electroporated cells was combined with 1 ml fresh medium and seeded in a 6-well plate. After 72 h, supernatant was filtered (0.45 µm) and added to naïve Huh7-Lunet CD81 cells, seeded the day prior at a density of $1.5 * 10^5$ cells in a 6-well plate. Cells were washed at 4 h post infection and at 72 h post infection, infected cells were harvested for luciferase assay as described above. All plasmids included in the experiment were subjected to Oxford Nanopore sequencing (Microsynth AG) prior to in vitro transcription to ensure that the integrity of the chimeric genomes was maintained.

## HCVpp

HIV-based particles bearing HCV envelope proteins (HCVpp) were used for studies of cell entry. For transfection, a mix of 2.16 µg pcDNA based envelope protein expression construct[83], 6.42 µg HIV gag-pol expression construct pCMVΔ8.74[84], 6.42 µg firefly luciferase transducing retroviral vector[85], 45 µl polyethyleneimine (PEI) and 740 µl Opti-MEM was created, vortexed for 10 s and incubated at RT for 20 min before being added to $1.2 * 10^6$ HEK293T cells which were seeded the

day prior in a 6 cm-diameter-dish. Medium was replaced after 6 h. 48 h after transfection, supernatant containing the pseudoparticles was passed through a 0.45 µm filter and used to infect $4 * 10^4$ naïve Huh7-Lunet CD81 cells seeded in a 12-well plate the day before. After 72 h, harvesting of samples and luciferase assay were performed as described above. To quantify HCVpp titers used for infection, SYBR Green based Product Enhanced Reverse Transcriptase assay (SG-PERT) was performed[86] using the Takyon SYBR green kit (Eurogentec).

## Sequence analysis

Sequence alignments were performed using ClustalOmega through the msa package[87] and visualised via Jalview (version 2.11.4)[88]. Based on 358 (gt1b)[21], 496 (gt1a), 509 (gt3a), 29 (gt4a) HCV amino acid sequences encompassing the whole viral polyprotein retrieved from ViPR, genotype specific consensus sequences were determined using EMBOSS cons[89]. The 358 gt1b sequences were also used to generate the phylogenetic tree of gt1b using the minimum evolution principle.

To analyse sequence cohorts, each patient's ISDR amino acid sequence was aligned to the genotype specific consensus sequence, and the number of differences was determined automatically through a script in R (version 4.5.1).

## Immunohistochemistry

Slides (thickness 2–3 µm) from 16 HCV infected human cases were examined. They were stained with haematoxylin and eosin (H&E) according to the standard protocol at the Institute of Pathology, Technical University Munich (TUM). Slides were then independently evaluated by two experienced liver and comparative pathologists. Immunohistochemistry (IHC) was performed using the Bond RXm autostainer system (Leica) with all reagents supplied by Leica. Tissue sections were first deparaffinised using the Leica de-wax kit to remove embedding medium, followed by rehydration through a graded alcohol series with decreasing ethanol concentrations (100%, 96%, 70%). Heat-induced epitope retrieval was performed using epitope retrieval solution 1 (H1), corresponding to a citrate buffer at pH = 6, for 30 minutes. Endogenous peroxidase activity was blocked with 3% H₂O₂. The tissue sections were incubated with the primary antibody (Hepatitis C virus immunostaining - HCV 9E10; 1:4000) at room temperature for 15 min. Antibody detection was carried out using the Polymer Refine Detection Kit. Visualization of antibody binding was achieved using the chromogenic substrate diaminobenzidine (DAB; Medac Diagnostica). Analysis was performed using Aperio ImageScope (Leica Biosystems, version 12.3). For each patient, in 6 regions of interest (500 × 500 µm) HCV positive cells were counted, regions of interest were selected to contain at least one HCV positive cell.

## Analysis of genetic determinants

Genetic determinants were analysed based on a previously described cohort[90]. Blood specimens collected at enrolment were used to generate host-genotyping data through the Affymetrix UK Biobank array, which covers ~820,000 markers. Quality control exclusion of SNPs using PLINK v1.9[91] consisted of: minor allele frequency (MAF) < 5%, call rate of <98 %, Hardy-Weinberg equilibrium $p < 10^{-5}$, batch difference and sex difference in allele frequency (SNPs with a significant different MAF, $p < 10^{-5}$, were filtered out). A total of 542,730 high-quality autosomal SNPs were retained for imputation. Individuals with a call rate <95% and had an outlier heterozygosity value (±3 SD from the same self-identified ancestry group's heterozygosity rate mean) were excluded. Sex of each remaining subject was verified by comparing the reported sex with the observed sex based on X chromosome method-of-moments F coefficient. Pairwise genotype concordance for all patients was assessed to identify duplicates. Patients with genotype concordance > 95% at a set of pruned SNPs were considered duplicates and the members of the pairs with the lower call rate were excluded.

Article

Identity-by-descent (IBD) analysis was done to detect duplicated samples. Imputation was performed on the TOPMed Imputation Server using the TOPMed reference panel version r2[92]. ~6.3 million of SNPs with an imputation quality score greater than 0.3 were retained for genetic analyses.

We conducted a principal component analysis (PCA) to evaluate population substructure and map self-identified racial/ethnic groups onto the estimated principal components (PCs). Kinship coefficients were estimated to select unrelated individuals using PC-Relate, as implemented in the R package GENESIS[93]. The SNPRelate package in R[94] was then used for PCA on pruned common genotyped SNPs and using a set of unrelated individuals, defined as pairwise kinship coefficients less than $2^{-9/2}$. These 2 estimation procedures were iterated to ensure that the kinship coefficients were unbiased in admixed individuals and that the PCs were computed over unrelated individuals. For the viral data, PCA was performed on the nucleotide data as follows. Tri- and quad-allelic sites were converted to binary variables, and the amino acid frequencies were standardised to have mean zero and unit variance. MATLAB (release 2015a, The MathWorks) was used to perform the PCA using the singular value decomposition function.

To test for association between human SNPs and HCV mutation count in ISDR region, we performed logistic regression using PLINK2[95], adjusted for the human population structure (ten first PCs) and the virus population structure (five first PCs). Age, gender, and virus sequencing centre were also included as covariates to control the potential confounding.

### Determination of TTV titers

TTV quantification was carried out using the TTV R-Gene assay (BioMérieux), a real-time PCR assay that targets the TTV 5'UTR. This assay has a dynamic range from 250 to $10^9$ copies/ml, with a detection limit of 250 copies/ml. TTV DNA was extracted from serum samples using the QIAsymphony SP platform (QIAGEN), and PCR was performed on a Light Cycler 480 Instrument II (Roche Diagnostics). The viral load was determined using a standard curve.

### Statistical analysis

Statistical analyses were performed using GraphPad Prism (Graphpad Software, version 8.4.3) or R (version 4.5.1). For numerical data, a two-tailed Student's t-test was applied when normal distribution was assumed; otherwise, the Mann-Whitney U test was utilised. For categorial values, Fisher's exact test was used.

### Ethics statement

This study was approved by the ethics committee of the medical faculty of Heidelberg University (ethics votes: S-399/2012, S-720/2022 and S-743/2023). 12 liver samples were provided by the tissue bank of the German Centre for Infection Research (DZIF, Heidelberg, Germany) in accordance with the regulations of the tissue bank and the approval of the ethics committee of Heidelberg University (ethics vote: S-399/2012). One liver sample of an HCV negative patient suffering from cholestatic liver disease was approved by the ethics committee of Klinikum rechts der Isar (MRI, Munich) (ethics vote: 518/19 S). The Montreal Hepatitis C Cohort study (HEPCO) is approved by the Research Ethics Committee of the Centre de Recherche du Centre Hospitalier de l'Université de Montréal (CRCHUM) (Approval number: SL 05.014). The studies were conducted in accordance with the local legislation and institutional requirements. The participants provided their written informed consent to participate in this study.

### Reporting summary

Further information on research design is available in the Nature Portfolio Reporting Summary linked to this article.

## Data availability

Parts of the FCH cohort were previously published under the accession numbers MK092096-MK092105 (FCH1-5) [https://www.ncbi.nlm.nih.gov/nuccore/?term=MK092096%3AMK092105+%5Bpacc%5D], MK092106-MK092111 (Non-FCH1-3) [https://www.ncbi.nlm.nih.gov/nuccore/?term=MK092106%3AMK092111++%5Bpacc%5D], OM222702 (GLT1) [https://www.ncbi.nlm.nih.gov/nuccore/OM222702], HQ719473 (BHCV1) [https://www.ncbi.nlm.nih.gov/nuccore/HQ719473] and JQ914274 (1a_FCH2) [https://www.ncbi.nlm.nih.gov/nuccore/JQ914274]. Sequence information on all other FCH patients generated during this study is deposited in GenBank under the following accessions: PV083181-PV083206 [https://www.ncbi.nlm.nih.gov/nuccore/?term=PV083181%3APV083206+%5Bpacc%5D], PX390011-PX390139 [https://www.ncbi.nlm.nih.gov/nuccore/?term=PX390011%3APX390139+%5Bpacc%5D]. NGS data from the GLT1 patient are deposited in the sequence read archive (SRA) under the BioProject accession PRJNA1214216. All other data generated during this study are included in this published article (and its supplementary information files). Source data are provided with this paper.

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

## Acknowledgements

We thank R. Klein, U. Herian and A. Hering for excellent technical assistance. For the generous gift of plasmids and antibodies, we are grateful to J. Bukh (H77) and C. Rice (S52, ED43, α-NS5A 9E10). The authors wish to acknowledge the role of HCV Research UK (funded by the Medical Research Foundation, award number C0365) in collecting and making available the data used in the generation of this publication. For the publication fee we acknowledge financial support by Heidelberg University. This work was funded by the Deutsche Forschungsgemeinschaft (DFG, German Research Foundation), Project-ID 519777725 to V.L. and 272983813 – TRR 179, to V.L., C.M., M.H. and R.T.; H.G.T.S. was supported by a stipend from the DZIF academy. X.F. and S.P.P. received funding from Instituto de Salud Carlos III, project PI22/00013, co-funded by the European Union. X.F. has received support from Secretaria d'Universitats i Recerca del Departament d'Economia i Coneixement (grant 2021_SGR_01322) and CERCA Programme/Generalitat de Catalunya. The Montreal cohort is supported by grants from the National Institutes of Health (NIH) U19AI159819 and the Canadian Institutes of Health Research (CIHR) (PJT-173467) to N.S. and J.B.; HCV sequence data in pregnancy were derived with support of the National Institute of Allergy and Infectious Diseases of the U.S. National Institutes of Health (R01AI096882) to J.H.

## Author contributions

P.R.: methodology, conceptualization, investigation, formal analysis, visualization, writing, review and editing; T.A.: investigation, review and editing; H.G.T.S.: investigation, review and editing; C.H.: investigation, review and editing; M.T.: methodology, investigation, review and editing; Z.W.: formal analysis, investigation, review and editing; C.F.: Investigation, review and editing; A.C.S.: investigation, review and editing; J.Q.: investigation, review and editing; H.C.: investigation, review and editing; M.R.: investigation, review and editing; L.B.: methodology, review and editing; J.H.: resources, review and editing; M.H.: resources, review and editing; R.T.: resources, review and editing; J.T.: resources, review and editing; STOPHCV: resources; P.S.: resources, review and editing; U.M.: resources, review and editing; N.H.S.: resources, review and editing; J.B.: resources, review and editing; C.R.: resources, review and editing; A.L.: resources, review and editing; R.A.B.: resources, review and editing; M.A.A.: supervision, resources, review and editing; C.M.: supervision, resources, review and editing; J.M.: resources, review and editing; X.F.: resources, review and editing; S.P.d.P.: resources, review and editing; V.L.: conceptualization, supervision, funding acquisition, writing, review and editing.

## Funding

## Competing interests

The authors declare no competing interests.

## Additional information

¹Department of Infectious Diseases, Molecular Virology, Section Virus-Host interactions, Heidelberg University, Heidelberg, Germany. ²German Center for Infection Research, partner site Heidelberg, Heidelberg, Germany. ³Institute of Pathology, School of Medicine and Health, Technical University of Munich, Munich, Germany. ⁴Nuffield Department of Medicine, Peter Medawar Building for Pathogen Research, University of Oxford, Oxford, UK. ⁵Chinese Academy of Medical Science Oxford Institute, University of Oxford, Oxford, UK. ⁶Department of Nephrology, University Hospital Heidelberg, Heidelberg, Germany. ⁷Department of Pediatrics, The Ohio State University College of Medicine, Columbus, OH, USA. ⁸Center for Vaccines and Immunity, The Abigail Wexner Research Institute, Nationwide Children's Hospital, Columbus, OH, USA. ⁹Department of Medicine II, Freiburg University Medical Center, Faculty of Medicine, University of Freiburg, Freiburg, Germany. ¹⁰Institute of Virology, Faculty of Medicine, University of Duesseldorf, Duesseldorf, Germany. ¹¹Department of Infectious Diseases Virology, University Hospital Heidelberg, Heidelberg, Germany. ¹²Department of Internal Medicine IV, University Hospital Heidelberg, Heidelberg, Germany. ¹³Centre de Recherche du Centre Hospitalier de l'Université de Montréal (CRCHUM), Montréal, QC, Canada. ¹⁴Département de Médecine, Université de Montréal, Montréal, QC, Canada. ¹⁵Département de médecine familiale, Université de Montréal, Montréal, QC, Canada. ¹⁶Faculty of Medicine, The Kirby Institute, University of New South Wales, Sydney, NSW, Australia. ¹⁷Faculty of Medicine, School of Biomedical Sciences, University of New South Wales, Sydney, NSW, Australia. ¹⁸MRC-University of Glasgow Centre for Virus Research, College of Medical, Veterinary and Life Sciences, University of Glasgow, Glasgow, UK. ¹⁹Liver Unit, Hospital Clínic de Barcelona, IDIBAPS, University of Barcelona, CIBEREHD, Barcelona, Spain. ✉e-mail: Volker.lohmann@med.uni-heidelberg.de

## STOPHCV investigators

Graham S. Cooke[20], Sarah Pett[21], Leanne McCabe[21], Chris Jones[20], Richard Gilson[21], Sumita Verma[22], Stephen D. Ryder[23], Jane D. Collier[24], Stephen T. Barclay[25], Aftab Ala[26], Sanjay Bhagani[27], Mark Nelson[28], Chin Lye Ch'ng[29], Ben Stone[30], Martin Wiselka[31], Daniel Forton[32], Stuart McPherson[33], Rachel Halford[34], Dung Nguyen[4], David Smith[4], M. Azim Ansari[4], Emily Dennis[21], Fleur Hudson[21], Eleanor J. Barnes[4,24] & Ann Sarah Walker[21]

[20]Imperial College London, London, UK. [21]University College London Medical School, London, UK. [22]Brighton and Sussex Medical School, Brighton, UK. [23]Nottingham University Hospitals NHS Trust, Nottingham, UK. [24]John Radcliffe Hospital, Oxford, UK. [25]Glasgow Royal Infirmary, Glasgow, UK. [26]University of Surrey, Guilford, UK. [27]Royal Free Hampstead NHS Trust Hospital, London, UK. [28]Chelsea & Westminster NHS Trust, London, UK. [29]Swansea Bay University Health Board, Swansea, UK. [30]Sheffield Teaching Hospitals NHS Foundation Trust, Sheffield, UK. [31]University Hospitals of Leicester NHS Trust, Leicester, UK. [32]St George's Hospital, London, UK. [33]Newcastle Upon Tyne Hospitals NHS Trust, Newcastle, UK. [34]Hepatitis C Trust, London, UK.

