## [Peer Review file · Nature Communications]

Highly replicating hepatitis C virus variants emerge in immunosuppressed patients causing severe disease

Corresponding Author: Professor Volker Lohmann

Version 0:

Reviewer comments:

Reviewer #1

(Remarks to the Author)

Dear Authors.

The manuscript entitled "Highly replicating hepatitis C virus variants emerge in immunosuppressed patients causing severe disease" claims to identify a replication-enhancing domain (ReED) in the NS5A protein that confers a higher replication fitness in clinical isolates. They go on further to show consensus sequences of accumulated mutations in the ReED replicate higher (in vitro) and were exclusively found in individuals with a more severe disease outcome, they found only 10% of viral genomes from large sequence cohorts showed the genetic signatures of high replicators (3 or more mutations in the ISDR region), but these were enriched in recipients of liver transplant, HIV co-infected individuals and patients developing HCC. The manuscript is very well written, structured well, with clearly presented, transparent figures, and relevant controls are used sufficiently. It is noteworthy that the authors have put a lot of detail in the work to make the results very compelling. The conclusions are very well supported.

However, overall the data confirms existing data that the accumulation mutations in the ISDR region can increase the replication fitness of the virus (eg. Kohashi et al J.Viral Hepat 2006), also that these accumulated mutations can also be associated with disease outcome (e.g. Hung et al, 2008 J viral hepatitis, El Shamy et al 2013, Hepatology). The extended region after the ISDR region that is included in the newly identified ReED did not seem to add more relevance to the previously known ISDR region (Fig 3A), therefore the research does not fully move beyond the current state-of-art.

Minor comments:

- It would be interesting to know if there are any key combinations of mutations (rather than just increased numbers) that were more common in the FCH pre- or post-, that could be markers of a more severe outcome, higher replication, and how these compare to other published work?
- In the samples from HIV coinfecting patients, were these patients immunosuppressed, undergoing therapy, with AIDS?

Reviewer #2

(Remarks to the Author)

This elegant work by Rothhaar et al., convincingly describes the identification of a region in the hepatitis C virus NS5A protein that correlates with increased replication in cell culture. The authors also convincingly associate this region to features of viral high replication in vivo, such as fibrosing cholestatic hepatitis. Moreover, changes in this region seem to be more prevalent in viral sequences from patients with immunosuppressive conditions. This is a novel and relevant study that sheds light on the pathogenesis of a very important viral disease. The amount and the quality of the functional assays is remarkable.

There are a few points that the authors could investigate further, or which need clarification.

1. Even though, without any doubts, this study has high relevance as it has uncovered an important association between viral sequence and viral pathogenesis in a specific manifestation of chronic HCV infection, these findings would have had much more relevance for the clinic 20 years ago. The authors mention in the discussion:

"Since HCV associated FCH currently mainly occurs in the context of HCV negative patients receiving an HCV positive organ, sequencing the viral quasispecies could allow an early identification of patients at risk."

This is true, however, the number of liver transplants with HCV positive organs is very small and the obvious course of action would be early DAA treatment. Nonetheless, the authors themselves mention a very clinically relevant aspect of their findings, also in the discussion:

“Our data thereby establish genome replication fitness as a new important variable in the in-terplay between HCV and the immune system potentially contributing to clearance or persis-tence of the infection, with further potential implications for DAA treatment and vaccine devel-opment.”

Therefore, it would be highly relevant to perform experiments to evaluate if highly replicative genomes respond differently to DAA implicated in pangenotypic regimens. In such experi-ments the authors could perform proof-of-concept treatment assays using the con1 and the GLT1 replicons. A suggested the experimental set-up would include treatment (dose-response assays) of con1 (low replication), GLT1 (high replication), con1-GLT1.ReED (rescue of high replication phenotype), and GLT1-con1.ReED (rescue of low replication phenotype). The au-thors should investigate NS5A inhibitors such as pangenotypic velpatasvir and pibrentasvir but also NS5B inhibitors such as sofosbuvir since the authors’ main hypothesis on the mechanism of action of the ReED also involve interactions with NS5B that might alter sofosbuvir suscep-tibility. For a more broad perspective, analysis of a DAA treatment cohort with known out-come could be considered.

2. Although the definition of the ReED is very convincing in some of the cases, such as in the GLT1, it would be relevant to investigate further if there are specific amino acids involved in the high replication phenotype. Therefore, replication experiments with the different mutations present in the ReED of the post-LTX sample compared to the pre-LTX samples for GLT1 would be highly relevant. Since the representation of the sequence information is not so clear, it is difficult to see how many amino acids do change after LTX, but it would appear from Fig.2C that GLT1 has between 5-7 changes compared to the two pre-LT populations. Exper-iments to prove the contribution of each of these changes or eventually combinations of these changes should be attainable. Refining the nature of the ReED could help to better identify whether an actual ReED is present or not. Currently the authors just compare the region to the consensus of the specific genotype which seems insufficient to accurately predict a ReED.

3. There is an obvious discrepancy between the pseudoparticle entry assays (Fig.S1B) and the virus infectivity assays (Fig.S1C-infectivity panel). Could this be because some of the GLT1-JFH1 chimeras might need cell culture adaptive mutations? The authors should se-quence the viruses and check that the GLT1 parts are conserved and have not changed if they need to reflect the properties of the patient sequences.

4. Please provide an actual amino-acid alignment of ReED sequences from the different samples of the GLT1 patient (like in fig.S3A).

5. The following statement in the results is highly questionable:

“For FCH patients with available complete replicase sequences, phylogenetic analysis showed no distinct FCH cluster indicating that most isolates could potentially convert into a high replicator and cause FCH by acquiring ReED mutations (Fig. S4B)”

First, the performed phylogenetic analysis (most likely any phylogenetic analysis) would not be able to cluster viral sequences according to FCH as seen in Fig.S4B. Non-clustering does not indicate that the isolates would be able to convert from low to high replicators. Of course, viral evolution could potentially drive any virus towards high replication, but it would depend for instance on how many changes in the ReED are needed to enhance replication.

6. Please, provide further clarification to the following statement in the discussion:

“FCH4&9 had a relatively high genetic diversity compared to the whole FCH cohort. Thus, divergent high replicator subtypes in the quasispecies might have remained unrecognised in the available consensus sequence.”

What does it mean that the FCH4 and FCH9 samples had high genetic diversity?

Do the authors mean that these two samples were genetically distant to the rest of consensus sequences from other patients?

Or do the authors refer to quasispecies diversity? In this case the authors must have access to the sequences within the quasispecies and would be able to see the variants which would allow the generation of other sequences for functional analysis, such as the master (more prevalent) sequence of the quasispecies, which from a functional point of view is more rele-vant than the consensus sequence in cases where the quasispecies diversity is very high.

Version 1:

Reviewer comments:

Reviewer #1

(Remarks to the Author)

Dear Authors,

For the revised version of the manuscript entitled “Highly replicating hepatitis C virus variants emerge in immunosuppressed patients causing severe disease” the authors have convincingly addressed the concerns raised.

I am happy with the current version and have no further comments to be addressed.

I would like to congratulate the authors on the excellent work.

Reviewer #2

(Remarks to the Author)

The authors have thoroughly addressed all my comments and concerns in a comprehensive manner. The revisions have significantly improved the clarity and relevance of the manuscript. I am satisfied with the responses provided and the changes made.

Reviewer #3

(Remarks to the Author)

Point by point response to reviewer comments

Reviewer #1 (Remarks to the Author):

Dear Authors.

The manuscript entitled “Highly replicating hepatitis C virus variants emerge in immunosuppressed patients causing severe disease” claims to identify a replication-enhancing domain (ReED) in the NS5A protein that confers a higher replication fitness in clinical isolates. They go on further to show consensus sequences of accumulated mutations in the ReED replicate higher (in vitro) and were exclusively found in individuals with a more severe disease outcome, they found only 10% of viral genomes from large sequence cohorts showed the genetic signatures of high replicators (3 or more mutations in the ISDR region), but these were enriched in recipients of liver transplant, HIV co-infected individuals and patients developing HCC.

The manuscript is very well written, structured well, with clearly presented, transparent figures, and relevant controls are used sufficiently. It is noteworthy that the authors have put a lot of detail in the work to make the results very compelling. The conclusions are very well supported.

We are very pleased about the highlighting of the scientific and editorial quality of our study by this reviewer.

However, overall the data confirms existing data that the accumulation mutations in the ISDR region can increase the replication fitness of the virus (eg. Kohashi et al J.Viral Hepat 2006), also that these accumulated mutations can also be associated with disease outcome (e.g. Hung et al, 2008 J viral hepatitis, El Shamy et al 2013, Hepatology). The extended region after the ISDR region that is included in the newly identified ReED did not seem to add more relevance to the previously known ISDR region (Fig 3A), therefore the research does not fully move beyond the current state-of-art.

We are aware of the excessive literature on the ISDR region, that has been published in the past. However, these studies have focused almost exclusively on gt1b and mainly found significant association between IFN treatment outcome and the number of ISDR mutations in Japan, but mostly lacking functional analyses, which was severely hampered at that time by the lack of suitable models. The above-mentioned study of Kohashi et al. was the only exception and indeed, is in good agreement with the new detailed analysis of mutations we added in response to the first comment of this reviewer below. Disease association was so far only indicated for HCC in the above cited papers, as well as by Giménez-Barcons, M., et al. Hepatology, 2001. However, the data were controversial, with Hung et al. showing an association of non-mutated ISDRs with HCC, whereas both other studies came to the same conclusion as we did. Therefore, our study contributes to a controversial discussion in the field. All these papers have been cited and discussed in the original version of the paper, except for Hung et al., which we unfortunately missed, but now has been added.

However, the main novelty of our paper is indeed not the identification of the ISDR, but the fact that high replicator variants are strongly associated with severe disease outcome after LTX, as shown by a highly significant association with FCH. The idea that immune suppression is the driving force of this process is supported by the correlation analysis showing increased abundance upon HIV co-infection and HCC.

We also show for the first time that replication enhancement by ISDR variants is found for all major genotypes across the world, by an extensive reverse genetics analysis going far beyond the existing literature. Here, we further added additional functional analyses of 2 gt1a ReEDs associated with FCH and 7 gt3a ReEDs from LTX patients (new Fig. 5A, C; Fig. S8A, B and S9A, B), to further extend on this novel aspect. Furthermore, selection of variants with increased RNA replication competence after LTX is shown here for the first time by a detailed functional analysis of dominating quasispecies prior and after LTX. So far, only one study suggested a selection for higher entry efficiency. We hope, also by the addition of the extensive novel data we added in response to both reviewers, that this reviewer agrees that our study contains sufficient novelty to be acceptable for publication.

Minor comments:

- It would be interesting to know if there are any key combinations of mutations (rather than just increased numbers) that were more common in the FCH pre- or post-, that could be markers of a more severe outcome, higher replication, and how these compare to other published work?

We have added an extensive mutational analysis to address this comment. First, to evaluate the importance of the C-term of the ReED, we tested ISDR variants in the context of a consensus gt1b C-term. Indeed, we show for two FCH variants that the ISDR chimeras have the same phenotype as the entire ReED. However, for GLT1, this is still not the case, arguing for our concept that the entire ReED should be included in functional studies (new Fig. 4A), while in silico analyses should focus on the number of ISDR mutations. We further phenotypically analyzed individual mutations (new Fig. 4B, C) and insertions (new Fig. 4E) in gt1b, and show that only single mutations at position P2209 substantially increased replication, as well as a small insertion found in two different FCH variants. We demonstrate that reversion of position 2209 to proline in the context of high replicator ReED indeed substantially reduced replication (new Fig. 4D). This data is widely in agreement with existing literature, but with a far higher magnitude of effects. We further show that a gt1b high replicator ReED enhances replication of gt1a (new Fig. S8C), arguing for conserved mechanisms across genotypes and added a functional analysis of frequent mutations in gt1a (new Fig. 5B, S8D) and gt3a ISDRs (new Fig. 5D, S9C), again with P2209L having the strongest phenotype. P2209 mutations were found enriched in LTX patients within the HCV research UK cohort. Still, we are reluctant to present mutations at this residue as a marker, since many other combinations of ISDR mutations conferred high replication and resulted in more severe disease lacking these mutations. In sum, accumulation of 3 or more mutations in the ISDR compared to the (sub)-genotype specific consensus sequence remains the best genetic marker of increased replication fitness.

- In the samples from HIV coinfecting patients, were these patients immunosuppressed, undergoing therapy, with AIDS?

Indeed, such information would be tremendously helpful, but, unfortunately, the HCV UK cohort only contains the status of HIV-co-infection, but no information beyond regarding HIV. However, this is of course a highly relevant question, which we are envisaging in future studies by approaching HIV-centric cohorts. We have addressed this important point in the discussion of the paper more intensively.

Reviewer #2 (Remarks to the Author):

This elegant work by Rothhaar et al., convincingly describes the identification of a region in the hepatitis C virus NS5A protein that correlates with increased replication in cell culture. The authors also convincingly associate this region to features of viral high replication in vivo, such as fibrosing cholestatic hepatitis. Moreover, changes in this region seem to be more prevalent in viral sequences from patients with immunosuppressive conditions. This is a novel and relevant study that sheds light on the pathogenesis of a very important viral disease. The amount and the quality of the functional assays is remarkable.

We are excited about the positive perception of the quality of our manuscript by this reviewer, in particular highlighting the novelty and data quality.

There are a few points that the authors could investigate further, or which need clarification.

1. Even though, without any doubts, this study has high relevance as it has uncovered an important association between viral sequence and viral pathogenesis in a specific manifestation of chronic HCV infection, these findings would have had much more relevance for the clinic 20 years ago. The authors mention in the discussion:

“Since HCV associated FCH currently mainly occurs in the context of HCV negative patients receiving an HCV positive organ, sequencing the viral quasispecies could allow an early identification of patients at risk.”

This is true, however, the number of liver transplants with HCV positive organs is very small and the obvious course of action would be early DAA treatment. Nonetheless, the authors themselves mention a very clinically relevant aspect of their findings, also in the discussion:

“Our data thereby establish genome replication fitness as a new important variable in the interplay between HCV and the immune system potentially contributing to clearance or persistence of the infection, with further potential implications for DAA treatment and vaccine development.”

Therefore, it would be highly relevant to perform experiments to evaluate if highly replicative genomes respond differently to DAA implicated in pangenotypic regimens. In such experiments the authors could perform proof-of-concept treatment assays using the con1 and the GLT1 replicons. A suggested the experimental set-up would include treatment (dose-response assays) of con1 (low replication), GLT1 (high replication), con1-GLT1.ReED (rescue of high replication phenotype), and GLT1-con1.ReED (rescue of low replication phenotype). The authors should investigate NS5A inhibitors such as pangenotypic velpatasvir and pibrentasvir but also NS5B inhibitors such as sofosbuvir since the authors' main hypothesis on the mechanism of action of the ReED also involve interactions with NS5B that might alter sofosbuvir susceptibility. For a more broad perspective, analysis of a DAA treatment cohort with known outcome could be considered.

We performed the experiments exactly as suggested by this reviewer and included three NS5A inhibitors (Daclatasvir, Pibrentasvir and Velpatasvir) as well as one NS5B

inhibitor (Sofosbuvir) in the analysis (new Fig. 3D, S6A, B). Overall, replication efficiency had no impact on the inhibitory capacity of both drug classes, suggesting that variations in replication fitness will not affect the efficiency of current treatment regimens. This is further supported by publications on DAA treatment of FCH patients, which we have cited in the discussion. We furthermore identified a post-LTX patient infected with an HCV isolate containing a high-replicator ReED (new Fig. S6C), who was efficiently cured by DAA treatment.

However, very surprisingly, we found that low NS5A inhibitor concentrations (10 pM for Daclatasvir, 1 pM for Pibrentasvir and Velpatasvir) increased replication of constructs with the Con1 low replicator ReED almost to the level of those with a high replicator ReED, thereby copying their phenotype. This was completely unexpected, but might point to the mode of action of high replicator ReEDs. We so far speculated that NS5A has an inhibitory function, restricting activity of the polymerase. Low concentrations of NS5A inhibitors therefore might inactivate NS5A, thereby releasing its inhibitory action on NS5B similarly to ReED mutations. We have added this interpretation to the discussion, albeit it is highly speculative. Anyways, we are grateful for suggesting this experiment.

We are furthermore currently reaching out to DAA treatment cohorts, and we appreciate this excellent suggestion. However due to excessive paperwork it was not possible to get access to such data within the timeframe of this revision.

2. Although the definition of the ReED is very convincing in some of the cases, such as in the GLT1, it would be relevant to investigate further if there are specific amino acids involved in the high replication phenotype. Therefore, replication experiments with the different mutations present in the ReED of the post-LTX sample compared to the pre-LTX samples for GLT1 would be highly relevant. Since the representation of the sequence information is not so clear, it is difficult to see how many amino acids do change after LTX, but it would appear from Fig.2C that GLT1 has between 5-7 changes compared to the two pre-LT populations. Experiments to prove the contribution of each of these changes or eventually combinations of these changes should be attainable. Refining the nature of the ReED could help to better identify whether an actual ReED is present or not. Currently the authors just compare the region to the consensus of the specific genotype which seems insufficient to accurately predict a ReED.

We have added an extensive mutational analysis to address this comment, not only based on sequence data upon evolution of GLT1, but on a broader scale of functional data available in this study. First, to evaluate the importance of the C-term of the ReED, we tested ISDR variants in the context of a consensus gt1b C-term. Indeed, we show for two FCH variants that the ISDR chimeras have the same phenotype as the entire ReED. However, for GLT1, this is still not the case, arguing for our concept that the entire ReED should be included in functional studies (new Fig. 4A), while in silico analyses should focus on the number of ISDR mutations. We further phenotypically analyzed individual mutations (new Fig. 4B, C) and insertions (new Fig. 4E) in gt1b, and show that only single mutations at position P2209 substantially increase replication, as well as a small insertion found in two different FCH variants. We demonstrate that reversion of position 2209 to proline in the context of high

replicator ReEDs indeed substantially reduced replication, confirming its important function in those ReEDs containing it (new Fig. 4D). We further show that a gt1b high replicator ReED enhanced replication of gt1a (new Fig. S8C), arguing for conserved mechanisms across genotypes. In addition, we added functional analyses of 2 gt1a ReEDs associated with FCH and 7 gt3a ReEDs from LTX patients (Fig. 5A, C; Fig. S8A, B and S9A, B), to allow a better definition of determinants of ReED mediated replication enhancement and added a functional analysis of frequent mutations in gt1a (new Fig. 5B, S8D) and gt3a ISDRs (new Fig. 5D, S9C), again with P2209L having the strongest phenotype. Despite these detailed studies, accumulation of 3 or more mutations in the ISDR compared to the (sub)-genotype specific consensus sequence remains the best genetic marker of increased replication fitness. In the light of the numerous high replicator ReEDs we have phenotypically characterized here, reaching high replication levels with very divergent combinations of mutations, it remains very challenging to get to a better definition. This might only change, if we can clarify the molecular mode of action of the ReED.

3. There is an obvious discrepancy between the pseudoparticle entry assays (Fig.S1B) and the virus infectivity assays (Fig.S1C-infectivity panel). Could this be because some of the GLT1-JFH1 chimeras might need cell culture adaptive mutations? The authors should sequence the viruses and check that the GLT1 parts are conserved and have not changed if they need to reflect the properties of the patient sequences.

Indeed, the pseudo particle model assessing entry competence is well established in the field and truly represents entry efficiency. However, assembly and release are the most difficult steps in the viral replication cycle to study, concerning the comparative analysis of variants, since any impact on RNA replication or entry will also apparently affect assembly and release. We used the best-established model in the field, based on fusing the gt2a JFH1 replicase to the various structural protein coding sequences of interest. However, several studies in the past have shown, that this approach requires adaptation in case of intergenotypic chimeras to be efficient. Therefore, the results obtained in such an approach need to be interpreted with care. However, there is no better alternative since we and many others have shown that the establishment of an isolate efficiently producing virus in cell culture requires numerous mutations (>20 for GLT1) for unknown reasons. Therefore, the assumption of this reviewer, that at least the loss of function of one of the chimeras might be due to incompatibility, requiring adaptive mutations, is very likely. However, since the system is based on transfection of a huge amount of in vitro transcribed genomes (>10¹²) and the limited time frame in absence of selective pressure, it is basically impossible that a phenotype in this assay can be based on the emergence of virus mutants.

To address this important point, we have described the experiment and its limitations in far greater detail in the results section of the revised manuscript. We have further performed whole-plasmid sequencing on all plasmid preparations used for in vitro transcription, to ensure that the integrity of the chimeras was not affected by the

amplification of the plasmid in bacteria and added this information to the materials and methods section.

4. Please provide an actual amino-acid alignment of ReED sequences from the different samples of the GLT1 patient (like in fig.S3A).

We are grateful for this helpful comment and have replaced the schematic in Fig. 2C by an alignment like in Fig. S3A.

5. The following statement in the results is highly questionable:

“For FCH patients with available complete replicase sequences, phylogenetic analysis showed no distinct FCH cluster indicating that most isolates could potentially convert into a high replicator and cause FCH by acquiring ReED mutations (Fig. S4B)”

First, the performed phylogenetic analysis (most likely any phylogenetic analysis) would not be able to cluster viral sequences according to FCH as seen in Fig.S4B. Non-clustering does not indicate that the isolates would be able to convert from low to high replicators. Of course, viral evolution could potentially drive any virus towards high replication, but it would depend for instance on how many changes in the ReED are needed to enhance replication.

We agree with the reviewer that our previous statement lacked clarity. The only aim of the phylogenetic analysis was to exclude that all FCH/high replicator variants would be found in a distinct subset of the gt1b tree. However, the wide distribution across the tree rather suggests that ReED mutations can drive replication fitness potentially in any HCV isolate. We have rephrased the statement and hope that it is now clearer for the reader.

6. Please, provide further clarification to the following statement in the discussion:

“FCH4&9 had a relatively high genetic diversity compared to the whole FCH cohort. Thus, divergent high replicator subtypes in the quasispecies might have remained unrecognised in the available consensus sequence.”

What does it mean that the FCH4 and FCH9 samples had high genetic diversity?

Do the authors mean that these two samples were genetically distant to the rest of consensus sequences from other patients?

Or do the authors refer to quasispecies diversity? In this case the authors must have access to the sequences within the quasispecies and would be able to see the variants which would allow the generation of other sequences for functional analysis, such as the master (more prevalent) sequence of the quasispecies, which from a functional point of view is more relevant than the consensus sequence in cases where the quasispecies diversity is very high.

This statement lacked an important piece of information and we are grateful for pointing at this issue. The claim on genetic diversity was based on available NGS data published by Gambato et al., 2019. However, this data was restricted to a part of NS5B, whereas the NS5A sequence data only reflect the consensus, by direct sequencing of PCR products. Therefore, in this divergent quasispecies found in the NGS data on NS5B, several true high replicator subspecies might be hidden, which we might have missed due to the lack of NGS data for NS5A. We have added this information to results and discussion and hope that thereby our statement gets clearer.